# Myxobacteria restrain *Phytophthora* invasion by scavenging thiamine in soybean rhizosphere via outer membrane vesicle-secreted thiaminase I

Chengyao Xia[1], Yuqiang Zhao[2], Lei Zhang[1], Xu Li[1], Yang Cheng[3], Dongming Wang[1], Changsheng Xu[1], Mengyi Qi[1], Jihong Wang[1], Xiangrui Guo[1], Xianfeng Ye[1], Yan Huang[1], Danyu Shen[3], Daolong Dou [3,4], Hui Cao [1], Zhoukun Li [1] ✉ & Zhongli Cui[1,4] ✉

Public metabolites such as vitamins play critical roles in maintaining the ecological functions of microbial community. However, the biochemical and physiological bases for fine-tuning of public metabolites in the microbiome remain poorly understood. Here, we examine the interactions between myxobacteria and *Phytophthora sojae*, an oomycete pathogen of soybean. We find that host plant and soil microbes complement *P. sojae*'s auxotrophy for thiamine. Whereas, myxobacteria inhibits *Phytophthora* growth by a thiaminase I CcThi1 secreted into extracellular environment via outer membrane vesicles (OMVs). CcThi1 scavenges the required thiamine and thus arrests the thiamine sharing behavior of *P. sojae* from the supplier, which interferes with amino acid metabolism and expression of pathogenic effectors, probably leading to impairment of *P. sojae* growth and pathogenicity. Moreover, myxobacteria and CcThi1 are highly effective in regulating the thiamine levels in soil, which is correlated with the incidence of soybean *Phytophthora* root rot. Our findings unravel a novel ecological tactic employed by myxobacteria to maintain the interspecific equilibrium in soil microbial community.

In natural communities, computational analysis of bacterial genomes revealed a network of universally distributed competitive and cooperative metabolic interactions[1]. Through intricate trophic interactions, microorganisms establish facilitation, syntrophy, or reciprocity relationships by complicated trophic interactions and thereby evolve social behaviors such as cooperation and cheating[2]. Specialized metabolic functions of keystone taxa might play essential roles in maintaining the stability of the soil microbiome[3]. In addition to competing for natural resources, community members modify their surrounding environment or influence the growth of species in their immediate vicinity by secreting metabolic intermediates or by-products[4]. Metabolic interactions play important roles in shaping the structure of microbial community structure, since the exchange of metabolites or metabolic by-products influences the stability of ecological functions and division of labor in microbial populations[5]. Trophic interactions in the holobiont are considered as the central

[1]Key Laboratory of Agricultural Environmental Microbiology, Ministry of Agriculture and Rural Affairs, College of Life Sciences, Nanjing Agricultural University, Nanjing 210095, China. [2]Institute of Botany, Jiangsu Province and Chinese Academy of Sciences, Nanjing 210014, China. [3]The Key Laboratory of Monitoring and Management of Plant Diseases and Insects of Ministry of Agriculture and Rural Affairs, College of Plant Protection, Nanjing Agriculture University, Nanjing 210095, China. [4]Academy for Advanced Interdisciplinary Studies, Nanjing Agricultural University, Nanjing 210095, China. ✉e-mail: zkl@njau.edu.cn; czl@njau.edu.cn

driver of microbial assembly[6]. Recent studies found that the wide distribution of amino acid auxotrophies promotes the interdependence of microbial community members[5] and increases the resilience to disturbance by antimicrobial agents[7]. Certain costly metabolites are recognized as public goods, produced by specific members of the community and shared by the whole community. Digestive enzymes, polymeric substances, chelating compounds, amino acids, and vitamins are supplied as public goods[8]. The exchange of public goods in clonal microorganisms with high genetic relatedness has been researched in detail[2].

Oomycetes, which are important pathogens of various organisms, are superficially resemble fungi but are classified as stramenopiles along with brown algae and diatoms[9]. The infamous oomycete genus *Phytophthora* causes devastating economic damage to crops, such as soybean root and stem rot caused by *Phytophthora sojae*[10]. Auxotrophy for vitamin B1 (thiamine) is well-known in *Phytophthora* pathogens[11]. As opposed to fungi, swimming zoospores of *Phytophthora* travel at speeds of 120–150 μm/s and migrate through soil water or surface to find and colonize new hosts attracted by host-specific signals[12,13]. During the migration and colonization, *Phytophthora* might exploit the essential vitamin B1 (thiamine) from the host or soil environment, establishing complex interactions through metabolite exchanges[11]. Genome sequence analysis has revealed that the majority of the sequenced microorganisms are auxotrophic, relying on other members of the communities to acquire necessary amino acids or vitamins[14]. B-vitamins are public goods in oceanic plankton communities[15]. Eukaryotic algae are frequently found to be vitamin B auxotrophs[16], and they form mutualistic relationships with vitamin-supplying bacteria[17]. However, the physiological and ecological significances of public thiamine-mediated trophic interactions in plant holobionts remains largely unexplored.

In terrestrial ecosystems, a growing green plant forms a holobiont with the microorganisms that grow on it. The plant microbiome comprises microorganisms with enormous phylogenetical diversity, including beneficials, pathogens, and neutrals from the aspect of plant health, whose assembly has adapted to the host root exudates[18]. Root exudates are rich in amino acids, vitamins, organic acids, and sugars[19], and provide public resources for the assembly and functional maintenance of the microbial community. The associated microorganisms determine the health, environmental fitness, growth, and development of the host plant[20,21]. Competition for essential nutrients between community members in the rhizosphere also determines the ecological function of the rhizosphere microbiome. The trophic network architecture of root-associated bacterial communities has been shown to determine invasion by *Ralstonia solanacearum* and plant health in tomato microcosms[22]. The beneficial root microbiota outcompeted the pathogens by secreting growth-inhibitory siderophores[23]. Exploring the thiamine supply and regulation in the holobiont of *Phytophthora*–plant–rhizosphere microbes has great significance and deserves further investigation.

Myxobacteria are fascinating and important prokaryotes with remarkable multicellular behaviors, which have recently been reclassified as phylum-level lineages according to a genome-based taxonomy[24]. Recent research shows that predatory myxobacteria interact with fungi[25] and bacteria[26,27], which are considered keystone taxa of natural microbial communities[28,29]. Myxobacteria appear in soil and plant rhizosphere with relatively high abundance[30], indicating a potential function in driving microbial interaction in the rhizosphere. Here, we describe an innovative mechanism by which myxobacteria inhibit the growth of *Phytophthora* and thereby prevent its invasion of soybeans. Myxobacteria release a novel thiaminase to scanvage the public thiamine in the soil, which in turn regulates the thiamine content and promotes plant protection. Our experimental findings highlight a unique trophic interaction by intervening in the availability of public goods in microbial communities. Understanding this special trophic interaction in the microbiome will be helpful in developing a new biocontrol strategy for *Phytophthora* diseases and designing resilient synthetic ecosystems in agriculture.

## Results

### Thiamine is available for *Phytophthora* growth both from soybean root exudates and rhizosphere bacteria

Numerous studies found that members of the genus *Phytophthora* are thiamine auxotroph[11]. Our results confirmed these observations that thiamine, thiamine monophosphate (TMP), and thiamine pyrophosphate (TPP) enable the mycelia growth of the three selected strains of *Phytophthora*. Whereas the thiamine precursors hydroxymethylpyrimidine (HMP) and hydroxyethylthiazole (HET) failed to restore *Phytophthora* growth (Supplementary Fig. 1a, b), which was consistent with previous reports[31]. Genomic analysis of the thiamine biosynthesis pathway referring to *Saccharomyces cerevisiae* revealed that *P. sojae* and *P. infestans* lack most of the thiamine synthesis genes, especially those involved in the synthesis of HMP and HET (Supplementary Fig. 1c). Homologs of *THI4* (thiazole synthase), *THI5* (HMP synthase), *THI6* (hydroxyethylthiazole kinase), and *THI20* (HMP kinase)[32] were not found in the genomes of *Phytophthora* species. By contrast, the *THI80* gene, which encodes a thiamine pyrophosphatase that converts thiamine or TMP into its active form TPP, is present in both *S. cerevisiae* and *Phytophthora*, enabling thiamine utilization[33]. Particularly, *THI6* absence leads to the inability to synthesize thiamine molecules from simpler precursors. These results indicate that *Phytophthora* only utilizes exogenous intact thiamine molecules for normal growth.

*P. sojae* is a biotrophic and saprophytic oomycete that causes devastating losses in soybean production due to root and stem rot[10]. During biotrophic growth, *Phytophthora* acquires thiamine from the host plants[9]. However, it is not clear how saprophytic *P. sojae* gets the thiamine source. It is well-known that thiamine can be synthesized by numerous bacteria, fungi, and plants[32]. To determine the presence of thiamine in plant root exudates and the effects on the growth of *Phytophthora*, three different varieties of soybeans (susceptible cultivar: Hefeng-47; resistant cultivars: Lvling-9 and Huning 95-1) were selected for the bioassay, and qPCR analysis was performed to qualify *P. sojae* P6497 biomass. Results showed that all three soybean root exudates enable the growth of P6497 in the defined medium without thiamine (Fig. 1a, b), and the biomass of P6497 increased by about one order of magnitude from the addition of root exudates ($p < 0.05$, Duncan's, Fig. 1b). The auxotrophic strain *Escherichia coli* K-12 Δ*thiE*, an organism able to grow only in the presence of exogenous thiamine, was used as the indicator strain in the bioassay. Growth restoration of *E. coli* K-12 Δ*thiE* by three soybean root exudates was also observed, as indicated by the increase in $OD_{600}$ values ($p < 0.05$, Duncan's, Supplementary Fig. 1d). To investigate the contribution of certain soil microorganisms as public goods in the plant microbiome, we recovered a total of 1267 bacterial isolates from rhizosphere soil sampled in four different fields. Among them, 65 isolates (5.13% of the total) demonstrated various degrees of thiamine production (Supplementary Table 6). According to 16 S ribosomal RNA gene sequencing, the isolates were distributed in 31 genera. Notably, *Escherichia* sp. T-1 was selected as a representative strain for further functional validation for its higher thiamine-producing ability. As expected, the culture supernatant of strain T-1 restored the growth of P6497 and strain K-12 Δ*thiE* in the defined medium without thiamine (Fig. 1a, b and Supplementary Fig. 1d). We then investigated the effect of strain T-1 on the growth of P6497 during co-cultivation. Adjacent inoculation of strain T-1 promoted the growth of P6497 (Fig. 1c, d). These results suggest that plants, as well as soil microorganisms, provide public thiamine for the growth of *P. sojae* in soybean rhizosphere soil (Fig. 1e).

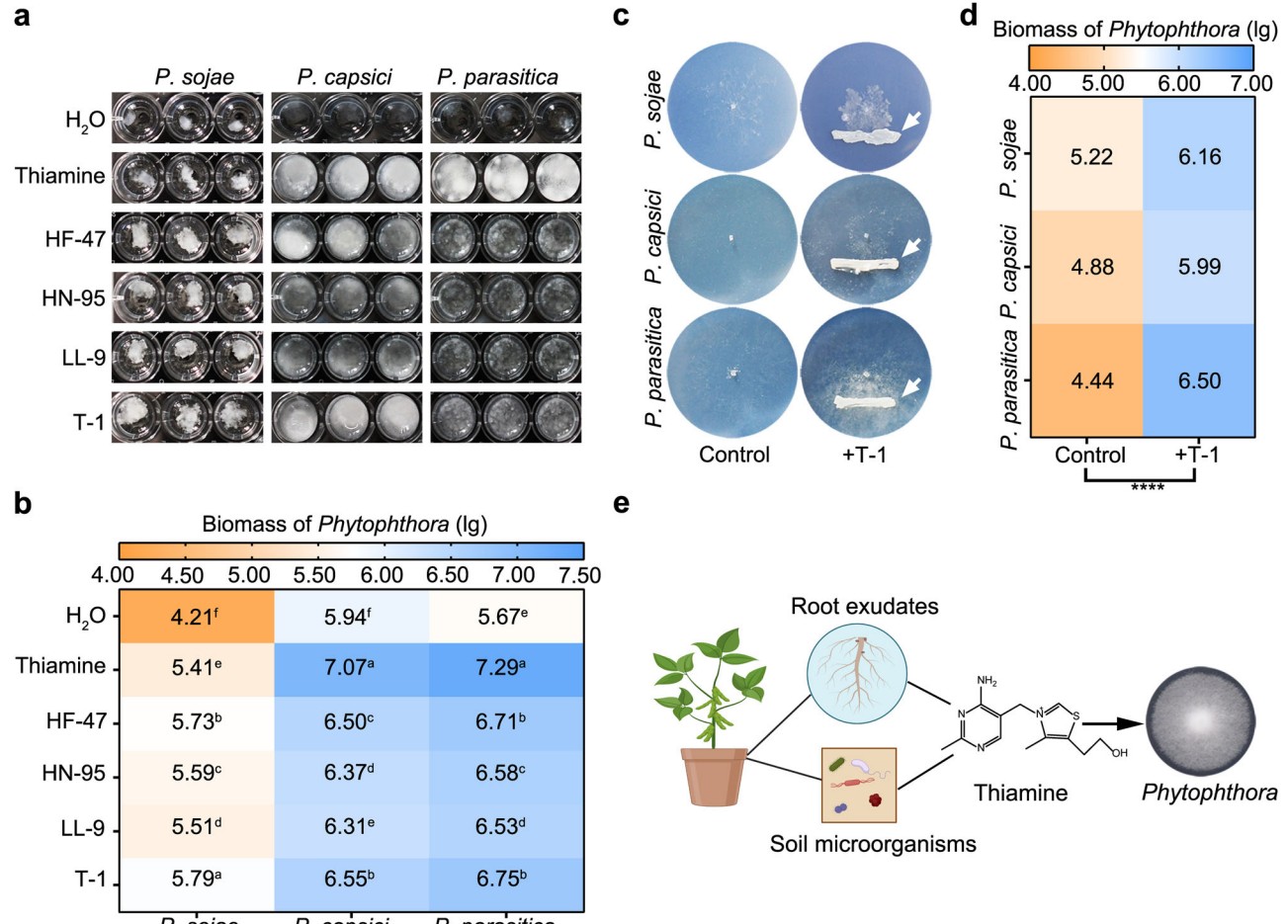

**Fig. 1 | Available thiamine from plant hosts and soil microorganisms promotes the growth of *Phytophthora*. a** Growth of *Phytophthora* with different additives. *P. sojae*, *P. capsici*, and *P. parasitica* were grown in liquid P1 medium with the addition of root exudates (1 mg/mL) from different soybean varieties and SUP (5 mg/mL) of strain T-1, respectively. Growth of *Phytophthora* was observed after 10 d of incubation at 25 °C using a stereomicroscope (Nikon SMZ-10). *Phytophthora* biomass in the liquid P1 medium from Fig. 1a was quantified by qPCR (**b**). The data represent the means ± SEM (*n* = 3 biological independent replicates). Means within columns followed by a different letter are significantly different (*p* < 0.05, one-way ANOVA, Duncan's multiple range test). **c** Co-cultures of *Phytophthora* and strain T-1 on solid P1 medium. A growth state was observed after 10 d of incubation at 25 °C. White arrows indicate the inoculation sites of strain T-1. **d** Quantitative analysis of the biomass of *Phytophthora* from the solid P1 medium by qPCR. HF-47, variety Hefeng-47; HN-95, variety Huning 95-1; LL-9, variety Lvling 9; T-1, thiamine-producing strain T-1 that isolated from soybean rhizosphere soil (Supplementary Table 6). The data represent the means ± SEM (*n* = 3 biological independent replicates). Asterisks indicate significant differences (****p < 0.0001, compared with control or H₂O group, two-sided, unpaired Student's *t*-test). **e** Schematic diagram of thiamine exchange between the soybean plant, rhizosphere microorganisms, and *Phytophthora* in the soil. Soil microorganisms and root exudates provide thiamine for the growth of auxotrophic *Phytophthora*. The cartoon image was drawn with Biorender. Source data are provided as a Source Data file.

## *Corallococcus* sp. EGB represses *P. sojae* growth by decomposing thiamine

In our previous research, we found that *Corallococcus* sp. EGB controls a wide range of pathogenic fungi as well as pathogenic bacteria by predation[25]. To investigate the interactions between myxobacteria and pathogenic oomycetes, we tested the action of strain EGB on the growth of P6497. Interestingly, strain EGB suppressed the growth of P6497 (Fig. 2a, b, GFP-labeled; Supplementary Fig. 2a, b, wild type) in a contact-independent manner in the plate confrontation assay, which was different from its contact-dependent predation on fungal and bacterial preys. Normal green fluorescence emission from the colony indicated the viability of the *Phytophthora* mycelia. Interaction between strain EGB and P6497 was further tested in liquid co-cultivation, and their biomass was quantified by qPCR. Results showed that liquid co-cultures of strain EGB resulted in 25-fold decrease of P6497 biomass (*p* < 0.05, Duncan's), but that of strain EGB remained unchanged (Fig. 2c, Supplementary Fig. 2c). Meantime, adhering of EGB cells to P6497 mycelia exhibited no destructive effects on the mycelial integrity (Supplementary Fig. 2d), but totally inhibited zoospores production (Supplementary Fig. 2e). These results indicate that strain EGB confines the growth of P6497 by inhibition rather than predation, and the adhesion of strain EGB results in decrease of P6497 biomass possibly by indirect physical hampering. To determine whether strain EGB inhibited the invasion of *P. sojae* in soybean plants, we conducted biocontrol experiments in a growth chamber. In vivo, pot culture (Supplementary Fig. 3) and in vitro leaf infection (Supplementary Fig. 6a, b) experiments showed that strain EGB could effectively control the infection of soybean plants by *P. sojae*.

Considering the inhibition interaction between strain EGB and P6497 in a contact-independent manner, we deduced that strain EGB represses *P. sojae* growth by secretory factors. To assess the underlying mechanism, morphological changes of P6497 mycelia co-cultured with lyophilized culture supernatant (SUPL) of strain EGB were investigated. Scanning electron microscopy (SEM) analysis showed that the mycelial structure remains intact after SUPL treatment (Supplementary Fig. 2d). The MTT assay showed that SUPL-treated mycelia reduce the indicator dye into formazan, and there was no significant difference in the formazan absorbance value between treatments (*p* = 0.8978, *t*-test),

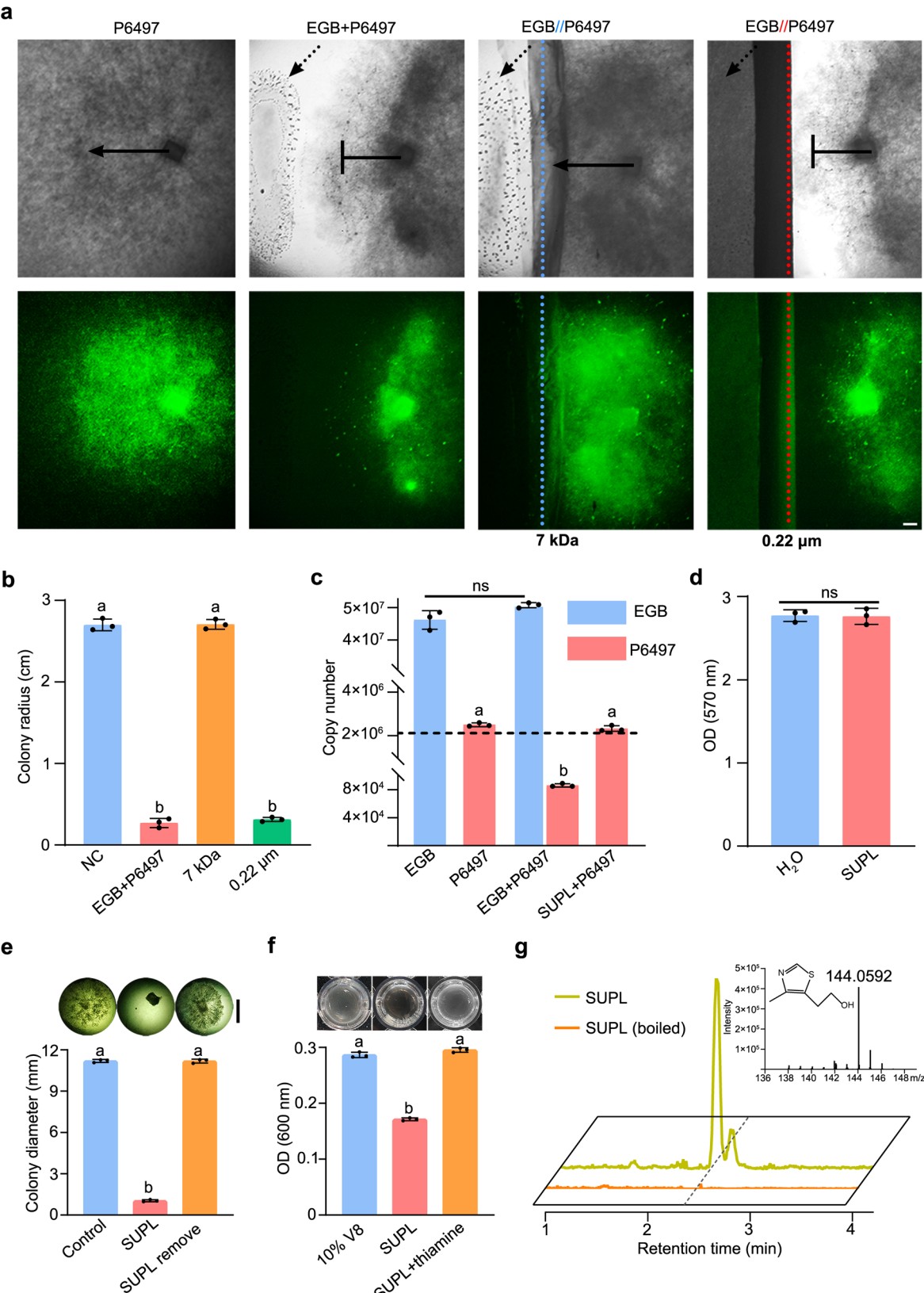

indicating that the growth-inhibited P6497 mycelia were still metabolically active (Fig. 2d, Supplementary Fig. 4a). Moreover, growth change in P6497 (Fig. 2c) or fluorescence quenching in GFP-labeled P6497 (Supplementary Fig. 4b) was not detected after SUPL treatment, which was consistent with the co-culture results (Fig. 2a). Remarkably, the mycelia could resume normal growth when SUPL-containing

medium was replaced with fresh 10% V8 medium (Fig. 2e). Hence, we concluded that strain EGB inhibits the growth of *Phytophthora* by a secretory factor in SUPL, and *Phytophthora* mycelia remain fully viable after treatments.

We hypothesized that strain EGB removes certain essential components required by *P. sojae* and limits mycelial development.

**Fig. 2 | *Corallococcus* sp. strain EGB inhibits the growth of *P. sojae* by decomposition of thiamine. a** Co-culture of strain EGB and GFP-labeled P6497 with (EGB// P6497) or without (EGB + P6497) membrane separation. The blue dotted line indicates the semi-permeable membrane (7 kDa molecular weight cutoff), the red dotted line indicates the membrane filter (0.22 μm pore size), and the dotted arrow indicates the location of strain EGB; → and ┝ indicates the normal or inhibited colony expansion, respectively. Scale bar, 1 mm. The images were recorded with a stereoscopic fluorescence microscope (Nikon, SMZ25, Japan). **b** Colony diameter measurement of GFP-labeled P6497 from the co-culture assay (**a**). The data represent the means ± SEM (*n* = 3 biological independent replicates). **c** Quantitative analysis of the biomass of strain EGB and P6497 from liquid co-culture by qPCR. The dotted line represents the initial inoculum of P6497. A growth state was also observed (Supplementary Fig. 2c). The data represent the means ± SEM (n = 3 biological independent replicates). **d** MTT staining of P6497 mycelia with and without SUPL treatment (Supplementary Fig. 4a). The metabolic activity was measured in the $OD_{570}$ (**d**). The data represent the means ± SEM (*n* = 3 biological independent replicates). **e** The growth of P6497 was observed after replacing the medium containing SUPL with fresh medium using a stereomicroscope (Nikon SMZ-10), and the colony diameter was measured. Scale bar, 5 mm. The data represent the means ± SEM (*n* = 3 biological independent replicates). **f** Growth of strain K-12 Δ*thiE* containing 10% V8 medium with or without SUPL. The addition of 2 μM thiamine recovered the growth. The growth of strain K-12 Δ*thiE* was determined by measuring the $OD_{600}$. The data represent the means ± SEM (*n* = 3 biological independent replicates). Means within columns followed by a different letter are significantly different in (**b**, **c**, **e**, **f** (*p* < 0.05, one-way ANOVA, Duncan's multiple range test), ns represents non-significant difference (two-sided, unpaired Student's *t*-test, *p* = 0.0583 for **c**, *p* = 0.8978 for **d**). **g** Identification of the thiamine decomposition products of SUPL or boiled SUPL treatment in 10% V8 medium by UPLC-MS. Source data are provided as a Source Data file.

Considering that *P. sojae* acquires the public thiamine from plants and soil microorganisms to facilitate self-growth and infection (Fig. 1e), we measured the content of thiamine in the medium treated with SUPL by bioassay with strain *E. coli* K-12 Δ*thiE* as the indicator. The $OD_{600}$ value of strain K-12 Δ*thiE* was reduced by half compared to the control group, implying a significant decrease of the thiamine content in the 10% V8 medium after SUPL treatment (*p* < 0.05, Duncan's, Fig. 2f). Meanwhile, we detected the thiamine-degradation product HET (mass-to-charge ratio of 144.048) by UPLC-MS analysis in the SUPL-treated V8 medium (Fig. 2g), but not for boiled-SUPL, indicating that addition of SUPL causes the decomposition of thiamine in the medium. In addition, biochemical properties and co-culture assays showed that SUPL fails to inhibit the growth of *Phytophthora* when the temperature and pH are up to 70 °C and pH 10.0, respectively (Supplementary Fig. 5a, b). These results imply that the strain EGB adopts a novel ecological tactic to interact with *Phytophthora* by secreting a thiamine-decomposing substance, i.e., thiamine decomposing enzyme, even though it exhibited extensive protease resistance (Supplementary Fig. 5a).

## Purification and characterization of the critical weapon from strain EGB to combat *P. sojae*

For identification of the protein involved in strain EGB inhibition, we developed a fast and efficient protocol to evaluate the inhibitory activity in 96-well plates with P6497 as the indicator to guide the protein purification process. The process to purify the protein from the supernatant failed though it showed significant inhibitory activity towards P6497. Unexpectedly, the majority of the activity was found in the cell membrane fraction (77.52%) rather than the soluble fraction of disrupted EGB cells, indicating that this active factor is a membrane protein (Supplementary Fig. 5a). Accordingly, we purified the protein from the membrane fraction by adding the detergent CHAPS, followed by ion exchange chromatography (IEC) on Hitrap capto Q, hydroxyapatite chromatography (HC) on Bio-scale™ mini CHT™, and gel filtration chromatography (GFC) on HiLoad 16/600 superdex 200 pg (Fig. 3a). Purification with EGB cells collected from a 5 L culture in LBS medium did not give the desired result, we increased the total culture volume to 50 L for purification. Based on the distribution of antagonistic activity in GFC fractions and the SDS-PAGE analysis of the purified protein, we determined that the target protein is found in a band around 45 kDa (Fig. 3b). Peptides generated by tryptic digestion were analyzed by mass spectrometry and the sequence of the most abundant fragment matched the protein encoded by the *EGBGL004693* gene (GenBank: WP_223633232.1) (Supplementary Table 5). EGBGL004693 was predicted to be a solute-binding protein in the NCBI database, while its thiaminase activity was predicted by conserved sequences alignment against known enzymes in the PDB database, which could decompose thiamine into pyrimidine and thiazole components in combination with a nucleophile. Hence, we designated EGBGL004693 as CcThi1 (with signal peptide).

The identified CcThi1 shared rather low sequence identity with known type-I thiaminases, including the prokaryotic and eukaryotic thiaminase I from *Paenibacillus thiaminolyticus* (PDB: 2THI) with 17.95% sequence identity[34], a thiaminase I from *Clostridium botulinum* (PDB: 4KYS)[35] with 19.18% sequence identity and a thiaminase I from *Naegleria gruberi* with 10.65% sequence identity (PDB: 4HCW)[36]. Phylogenetic tree analysis showed that members of thiaminase I and II families are divided into two branches. Notably, thiaminase I enzymes from myxobacteria formed a separate branch from those of other species (Supplementary Fig. 5c). However, the evolutionary relationship of thiaminase I homologs was closer in taxa within myxobacteria, as CcThi1 shared 70.15% sequence identity with MXAN_4523 (GenBank: ABF86831.1) from the model strain *Myxococcus xanthus* DK1622, which was regarded as homolog of CcThi1. Signal peptide prediction showed that CcThi1 contains a type II lipoprotein signal peptide consistent with its localization on the cell membrane of strain EGB.

CcThi1 was then heterologously expressed in *E. coli* BL21 (DE3) with (SP-CcThi1) or without (CcThi1) the signal peptide. We verified the decomposition and inhibitory activities of both recombinant proteins against P6497, indicating that lipoation and membrane association are not necessary for its growth-inhibitory effect against *Phytophthora* (Supplementary Fig. 5d). Thus, CcThi1 was selected for the following tests. In the bioassay, 0.12 nM CcThi1 exhibited excellent ability to suppress mycelial growth and effectively reduced the production of aerial mycelia. Zoospores are essential for the infection of plants by *P. sojae*[10], and 0.12 nM CcThi1 restricted germ tube elongation of zoospores, reducing their ability to form colonies. Nevertheless, even a high concentration of CcThi1 (2.4 nM) still failed to inhibit the germination of zoospores (Fig. 3c). We measured the content of thiamine in zoospores and speculated that the thiamine stored within the spores (1.3 amol/spore) is sufficient for the germination of zoospores (Supplementary Fig. 5e). Furthermore, CcThi1 effectively limited the infection of soybean leaves by P6497 in the biocontrol assay. The diameter of disease spots was significantly reduced with the increase of CcThi1 concentration (*p* < 0.05, Duncan's, Supplementary Fig. 6a, b).

A nucleophile (Nu⁻) is essential for the decomposing of thiamine into a pyrimidine component and thiazole component by thiaminase I (Fig. 3d). With 4-nitrothiophenol (4-NTP) as the nucleophile, CcThi1 exhibited a specific activity of 1.76 U/mg and $K_m$ value of 22.3 μM for the decomposition of thiamine (Supplementary Fig. 5d). At the same time, CcThi1 could effectively degrade thiamine without the presence of a nucleophile (Fig. 3e and Supplementary Fig. 6c, d), which is consistent with a previous report that thiamine itself opens the thiazole ring and participates as a nucleophile in the reaction[37]. It is worth mentioning that CcThi1 is the first thiaminase I enzyme functionally identified in myxobacteria with the ability to combat *Phytophthora*.

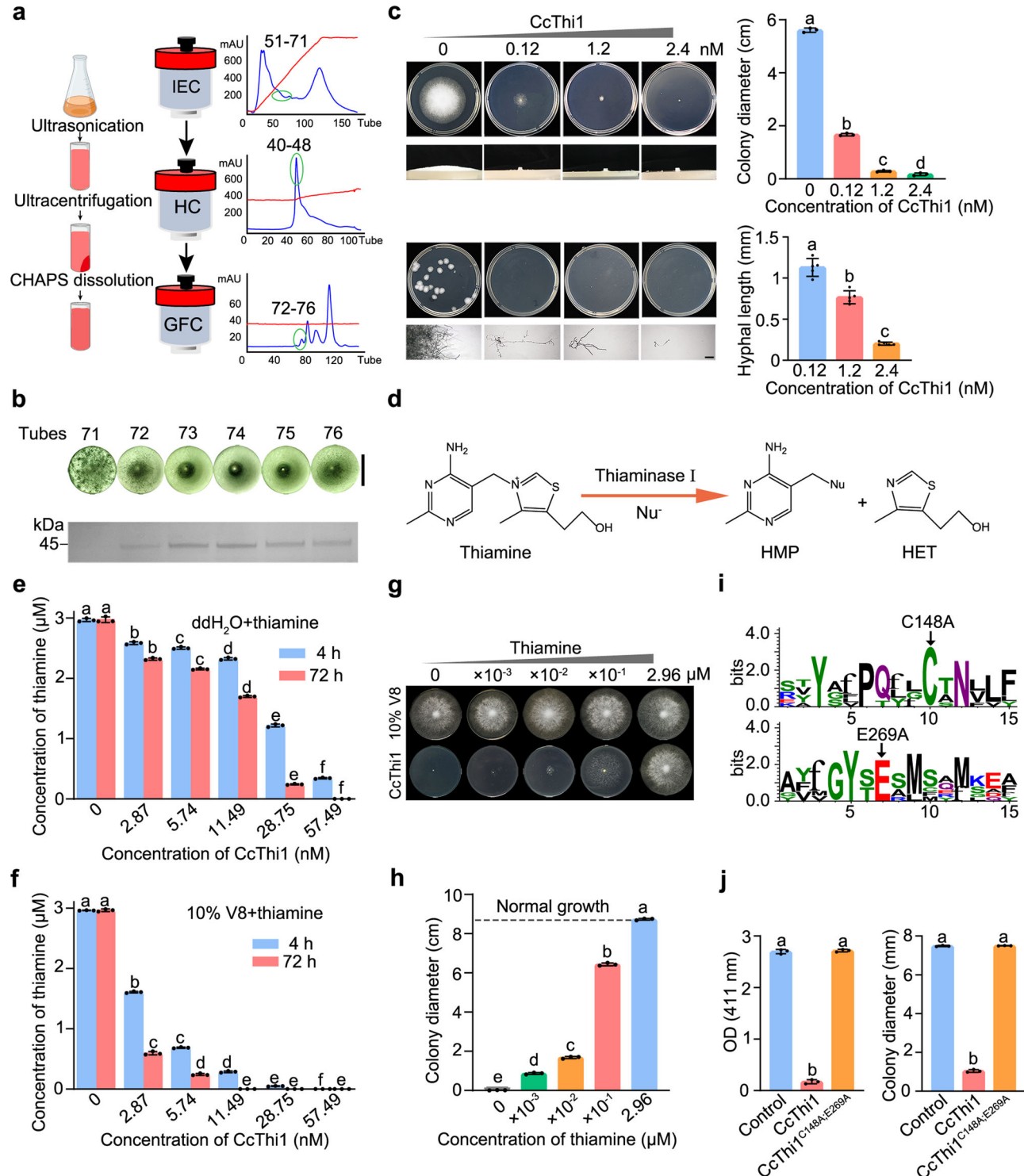

## CcThi1 limits the growth of *P. sojae* by decomposing essential thiamine

Based on the biochemical characteristics of CcThi1 (Fig. 3d), we further investigated the decomposition process. Thiamine was added to the medium at a final concentration of 2.9 µM, and its residual concentration was measured using the potassium ferricyanide method (detection limit: 0.16 µM). Dose-dependent degradation of thiamine by CcThi1 was detected in 10% V8 medium (Fig. 3f), which was more rapid than in pure water (Fig. 3e). Thiamine content was undetectable when incubated with 29 nM CcThi1 for 4 h, and *P. sojae* could no longer grow in 10% V8 medium after it was treated

with CcThi1 (data not shown). However, when additional thiamine was added to the medium along with 0.24 nM CcThi1, mycelial growth and germ tube elongation of zoospores recovered, and this effect was dependent on the dose, with 2.96 µM thiamine completely restoring the growth of P6497 in the presence of CcThi1 (Fig. 3g, h, Supplementary Fig 7a, b). This restoration was also observed from the in vitro soybean leaf infection assay with supplementation of thiamine (Supplementary Fig 7c, d). Above results show that CcThi1 restricts the mycelial growth of *Phytophthora* by decomposing the thiamine in the medium, which is essential for its growth and plant infection.

**Fig. 3 | Thiaminase I CcThi1 from the cell membrane of strain EGB inhibits *P. sojae* growth by decomposition of essential thiamine. a** Schematic diagram of the purification process with corresponding gel chromatograms. IEC ion exchange chromatography, HC hydroxyapatite chromatography, GFC gel filtration chromatography. Blue curve, protein concentration; red curve, salt concentration; green circle, active range. The cartoon image was drawn with Biorender. **b** Growth inhibition by the purified proteins from GFC and the corresponding SDS-PAGE electropherograms. The P6497 growth was observed using a stereomicroscope (Nikon SMZ-10). Scale bar, 5 mm. **c** Inhibition of mycelial growth by various concentrations of CcThi1. The aerial mycelia of P6497 were observed, and the growth inhibition was assessed by measuring the colony diameter (up). The data represent the means ± SEM (*n* = 3 biological independent replicates). CcThi1 inhibited the elongation of zoospore germ tubes of P6497. Microscopic observation of mycelia after CcThi1 treatment was performed to measure the hyphal length (down). Scale bars: 100 μm. The data represent the means ± SEM (*n* = 5 biological independent replicates). **d** Diagram of the reaction catalyzed by thiaminase I. Nu⁻ represents a nucleophile. **e, f** Measurement of thiamine content in the reaction system containing thiamine and $H_2O$ (**e**) or 10% V8 medium (**f**) with different concentrations of CcThi1. The data represent the means ± SEM (*n* = 3 biological independent replicates). **g, h** Addition of thiamine restored the CcThi1-mediated growth inhibition of P6497 (**g**), and the growth state was assessed by measuring the colony diameter (**h**). The data represent the means ± SEM (*n* = 3 biological independent replicates). **i** WebLogo depicting conserved residues of CcThi1. The black arrows indicate the predicted critical active-site residues C148 and E269. **j** Thiaminase I activity and growth inhibitory effect of CcThi1 and the mutant CcThi1^C148A; E269A (Supplementary Fig. 11). The data represent the means ± SEM (*n* = 3 biological independent replicates). Means within columns followed by different letters are significantly different in **c, e, f, h, j** (*p* < 0.05, one-way ANOVA, Duncan's multiple range test). Source data are provided as a Source Data file.

To further confirm that thiamine is the target of CcThi1, we selected the Δ*thiE* mutant of *E. coli* K-12 and the Δ*thi6* mutant of *Saccharomyces cerevisiae* BY4743, which both cannot synthesize thiamine from simple precursors. As anticipated, treatment of thiamine-containing medium with 2 μM CcThi1 significantly restricted the growth of strains K-12 Δ*thiE* and BY4743 Δ*thi6* (*p* < 0.05, Duncan's, Supplementary Fig. 8). In addition, CcThi1 exhibited a clear antagonistic effect against other species of *Phytophthora* in a dose-dependent manner (Supplementary Fig. 9). Antimicrobial activity assays were conducted with 4 bacteria (including Gram-negative bacteria *E. coli*, *Pseudomonas syringae*, and *Dickeya solani*, as well as the Gram-positive bacterium *Bacillus subtilis*) and 6 fungi (filamentous fungi *Magnaporthe oryzae*, *Fusarium oxysporum*, *Botrytis cinerea*, *Fusarium graminearum*, and *Trichoderma harzianum*, as well as the yeast *Pichia pastoris*). Since most of the tested fungi and bacteria are thiamine prototrophs, CcThi1 showed no antimicrobial effects against these microorganisms (Supplementary Fig. 10); Also, it was not effective against the fungus *Magnaporthe oryzae* and the oomycetes *Pythium oligandrum*, which can salvage the products of enzymatic reactions for thiamine synthesis.

The critical catalytic residues of thiaminase I are highly conserved, and the active site cysteine (C) is activated by an adjacent glutamate residue (E)[35]. Multiple sequence alignment revealed that the corresponding active-site residues of CcThi1 are C148 and E269 (Fig. 3i). To determine their contribution to catalytic activity, we converted them to alanine (A) by site-directed mutagenesis. The resulting mutant CcThi1^C148A; E269A completely lost its thiamine decomposition ability and had no inhibitory effect against P6497 (Fig. 3j, Supplementary Fig. 11). These results indicate that the thiamine decomposition activity of CcThi1 is essential for the growth inhibition of *Phytophthora*.

To further explore the metabolic changes of *Phytophthora* at the cellular level, the transcriptome of P6497 treated with CcThi1 was analyzed. A total of 2186 genes exhibited significantly changed expression levels (*q* ≤ 0.05) in CcThi1-treated P6497 compared with the control, 1044 of which were downregulated (fold change < −2), while 1142 were upregulated (fold change > 2) (Supplementary Fig. 12a). As the active form of thiamine, TPP is involved in various crucial pathways of carbohydrate and amino acid metabolism, including glycolysis, the citric acid cycle, the pentose phosphate pathway, as well as the synthesis of the branched-chain amino acids valine, leucine and isoleucine[38]. KEGG pathway analysis showed that genes involved in amino acid metabolism, lipid metabolism, and carbohydrate metabolism were greatly affected (*P* ≤ 0.05), which is in agreement with their thiamine requirements (Supplementary Fig. 12b). Analysis of the valine, leucine, and isoleucine synthesis pathway (Pathway ID: Ko00290), as well as the degradation pathway (Pathway ID: Ko00280), showed that 66.67% of the relevant genes in the synthesis pathway were significantly downregulated, while 75% of the genes in the degradation pathway were significantly upregulated (Supplementary Fig. 12c). Interestingly, we found that treatment with CcThi significantly affected the expression of the CRN (crinkling and necrosis proteins), RXLR, NLP (NEP1-Like Family), and elictin-like effectors[39] and virulence factors[40] of *P. sojae*, which are important contributors to plant pathogenesis. Among them, the virulence factor pectate lyase PL1[40] and the effector CRN showed significant downregulation (Supplementary Fig. 12d), indicating that thiamine deficiency from CcThi1 decomposition interferes with amino acid metabolism and expression of pathogenic effector, probably leading to the growth inhibition and pathogenic infection of *Phytophthora*.

## Extracellular secretion of CcThi1 is mediated by outer membrane vesicles

Mature lipoproteins are transported to the outer membrane via the Lol (localization of lipoproteins) system, while a specific residue at the +2 position of the signal peptidase cleavage site (e.g., aspartic acid) often results in inner membrane retention[41]. Since CcThi1 is a lipoprotein located on the cell membrane and does not possess inner membrane retention signals, we were interested in exploring the choice of export channel during interaction with *Phytophthora*. Outer membrane vesicles (OMVs) are nanoparticles that bacteria shed from their membrane[42]. It has been demonstrated that OMVs generated by myxobacteria function in both inter-colony communication and as predatory weapons against other bacteria[43]. These observations prompted us to investigate whether the extracellular activity of CcThi1 is associated with OMVs.

We collected the cell-free culture supernatant (SUP) of strain EGB and harvested OMVs fractions by ultracentrifugation. The ability of the fractions to inhibit the growth of P6497 was determined, and the results showed that the active components in the supernatant were concentrated in the OMVs, while the activity of the OMV-free SUP (SUPU) was significantly reduced by 90% (*p* < 0.05, Duncan's, Fig. 4a and Supplementary Fig. 13). Subsequently, we measured the catalytic activity of the fractions toward thiamine using 4-NTP as a nucleophile. Compared with SUP and SUPU, 85.6% of the enzymatic activity was concentrated in OMVs (Fig. 4b), and the localization of CcThi1 to the membrane and OMV fractions was further confirmed by western blot analysis (Fig. 4c, d).

Since it is difficult to genetically manipulate strain EGB, we deleted the *ccthi1* homolog *mxthi1* (*MXAN_4523*) in the model strain *M. xanthus* DK1622, generating CL1003, which almost completely lost its antagonistic effect against P6497. Complementation of CL1003 with *ccthi1* and *mxthi1* resulted in the CL1004 (*ccthi1* from strain EGB) and CL1005 (*mxthi1* from DK1622), which fully restored the antagonistic effects against P6497 (Fig. 4e, f). Similar to strain EGB, the extracellular antagonistic and thiamine decomposition activities of DK1622 were mainly detected in OMVs (90%). MxThi1 was not detected in CL003 OMVs by western blot analysis (Fig. 4a–c), while negligible antagonistic activity remained in the OMVs of CL1003 at the same protein

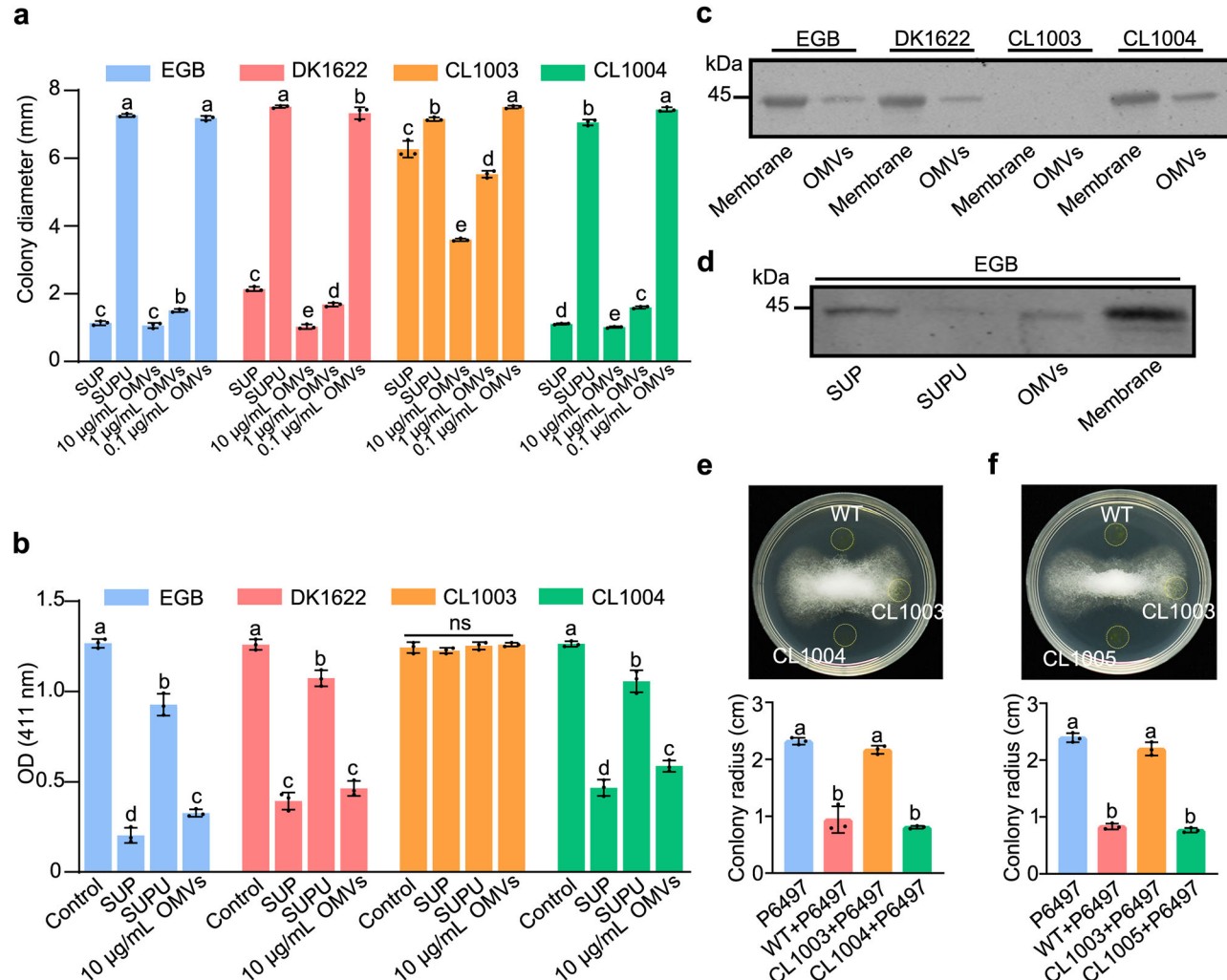

**Fig. 4 | OMVs are involved in the extracellular secretion of membrane-bound CcThi1 and its homolog from the model strain *M. xanthus* DK1622.**
**a**, **b** Determination of the growth inhibition ability (**a**) and thiaminase I activity (**b**) of OMVs from different myxobacterial strains. The SUP and SUPU fractions were used as control. The growth inhibition assay was performed in 96-well plates. The growth of P6497 was observed using a stereomicroscope (Nikon SMZ-10) (Supplementary Fig. 13), and the colony diameter was statistically analyzed. The thiaminase I activity of SUP, SUPU, and OMVs was determined by measuring the absorbance at 411 nm[63] (200 μL components and 200 μL reaction buffer at 37 °C for 1 h). The data represent the means ± SEM (*n* = 3 biological independent replicates).

**c**, **d** Western blot analysis of CcThi1 and its homolog in the membrane fraction, OMVs (**c**), SUP, and SUPU (**d**) using the anti-CcThi1 antibody. **e**, **f** Co-culture assay of *Myxococcus* strains with P6497 on 10% V8 agar plates. DK1622 and its derivative strains (5 μL, 10[7] cells/mL) were inoculated next to colonies of P6497. The growth of P6497 was observed after 7 d, and the colony diameter was measured. CL1003, Δ*MAXN_4523*; CL1004, Δ*MXAN_4523::ccthi1*, Kan[r]; CL1005, Δ*MXAN_4523::MXAN_4523*, Kan[r]. The data represent the means ± SEM (*n* = 3 biological independent replicates). Means within columns followed by different letters are significantly different in (**a**, **b**, **e**, **f**) (*p* < 0.05, one-way ANOVA, Duncan's multiple range test). ns represents the non-significant difference in (**b**). Source data are provided as a Source Data file.

concentration (10 μg/mL), and this low residual activity might be attributed to other components with anti-oomycete activity (Supplementary Fig. 14).

**Myxobacteria control soybean root rot by regulating public thiamine levels in rhizosphere soil**
Since CcThi1 exhibited an excellent antagonistic effect against P6497, we designed a greenhouse pot experiment to investigate the role of myxobacterial thiaminase in the soybean rhizosphere. To mimic the natural infestation process, we used the inoculum layer method to infect soybean plants[44]. As anticipated, soybean plants (susceptible cultivar hefeng-47) were seriously injured by P6497. Inoculation of strain EGB and DK1622 effectively controlled the *Phytophthora* root rot disease with efficiency of 50.34% and 46.23%, respectively. In the meantime, direct treatment with CcThi1 (1.2 nmol/pot every other day) could control the disease with 40.31% control efficiency. By contrast, the control efficiency of the *ccthi1* mutant strain CL1003 was reduced

by approximately 40% compared to DK1622 (26.65% vs. 46.23%). Moreover, the *ccthi1* complementation strain CL1004 recovered its biocontrol performance to a level comparable to DK1622 (44.34%) (Fig. 5a, b and Supplementary Fig. 15).

Based on these results, we speculated that the occurrence of the disease may be related to the thiamine content in the root ecosystem. We first collected the root–soil at day 21 for DNA extraction to analyze the changes in the abundance of P6497 between the different groups by qPCR. The abundance of P6497 was reduced to varying degrees by all treatments. Treatment with strain EGB had the most significant effect (*p* < 0.05, Duncan's), with a remarkable 45-fold reduction in the abundance of P6497, followed by CcThi1, strains DK1622 and CL1004 treatments with an approximately 4.5-fold reduction, while strain CL1003 resulted in a 3-fold reduction (Fig. 5c). Subsequently, we collected rhizosphere soil for thiamine extraction on days 7, 14, and 21. Since the thiamine content in the soil was too low to be quantified by HPLC (thiamine detection limit of 0.13 μM), we monitored the

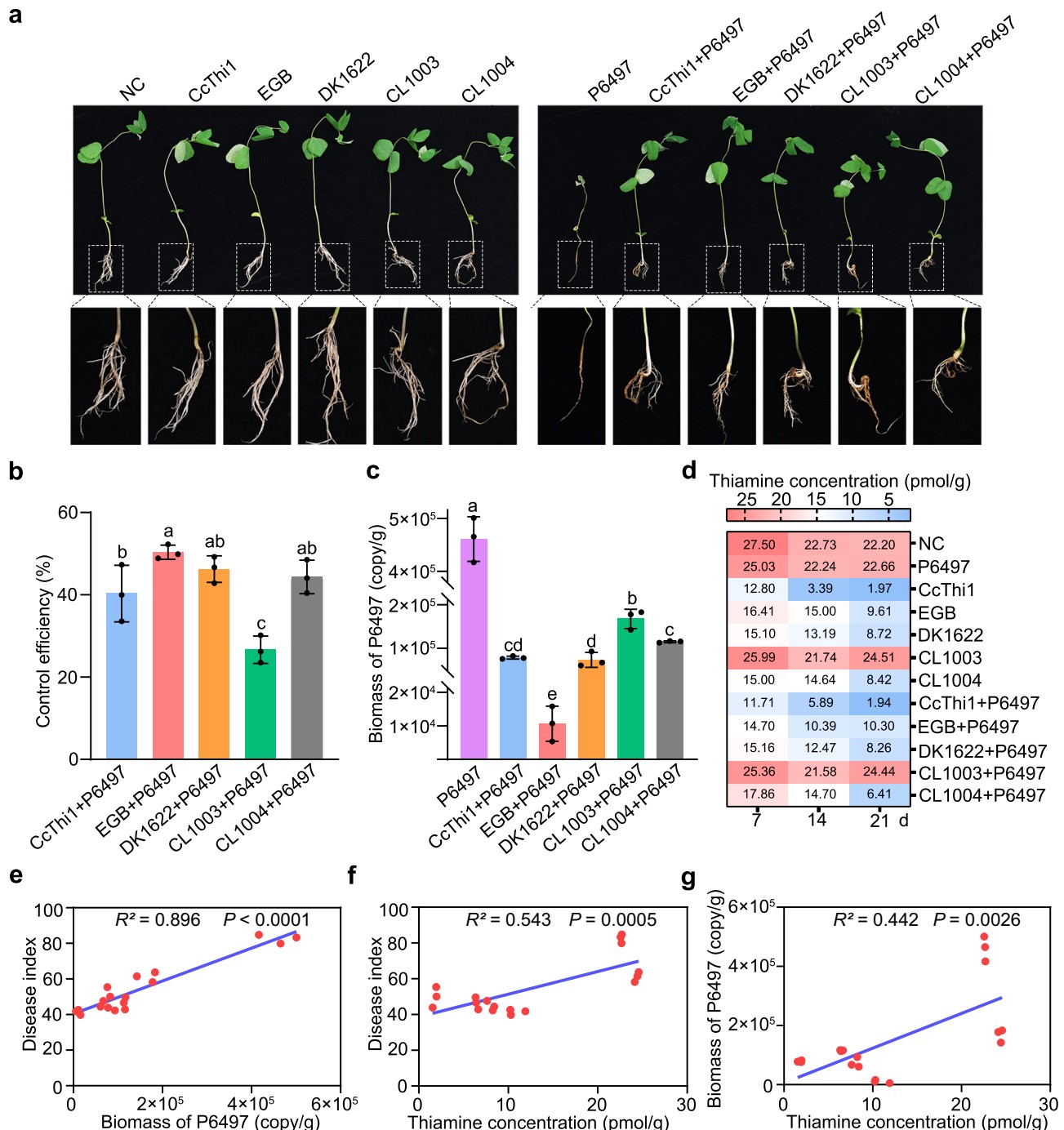

**Fig. 5 | Myxobacteria and secreted CcThi1 control *Phytophthora* root rot of soybean by decreasing the availability of thiamine in the soil. a** Effects of myxobacteria and CcThi1 on soybean root rot caused by *Phytophthora*. Soybean seedlings from different treatments were shown in Supplementary Fig. 15a, and the corresponding symptoms of *Phytophthora* root rot were shown below. **b** The biocontrol activity of myxobacterial strains and CcThi1 against *Phytophthora* root rot of soybean. Water was used as the untreated control. The data represent the means ± SEM ($n = 3$ biological independent replicates). **c** The abundance of P6497 in rhizosphere soil of soybean seedlings from different treatments after 21 d of co-culture under greenhouse conditions. The data represent the means ± SEM ($n = 3$ biological independent replicates). Means within columns followed by different letters are significantly different in (**b**, **c**) ($p < 0.05$, one-way ANOVA, Duncan's multiple range test). **d** Determination of thiamine content in soybean rhizosphere soil from the above pot experiment at day 7, 14, and 21, respectively. The concentration of thiamine (pmol/g) was indicated with different colors. **e**–**g** Positive correlations between the biomass of P6497 and disease index ($p < 0.0001$) (**e**), the thiamine concentration (day 21) and disease index ($p = 0.0005$) (**f**), the thiamine concentration (day 21) and P6497 biomass ($p = 0.0026$) (**g**). Correlation analysis was performed using two-sided Pearson's *r* correlation in GraphPad Prism 8.0.2. The data represent the means ± SEM ($n = 3$). Source data are provided as a Source Data file.

changes in thiamine content using bioassay. Treatments with strains EGB, DK1622, and the complementation strain CL1004 significantly reduced the thiamine levels in the soil by 54.55%, 63.55%, and 71.71% compared to the control group on day 21, respectively. By contrast, the

application of strain CL1003 had no significant effect on the thiamine concentration in the soil, which fluctuated slightly but remained generally stable over 21 d, in line with the P6497 and control groups. In addition, the thiamine content in the group with the addition of

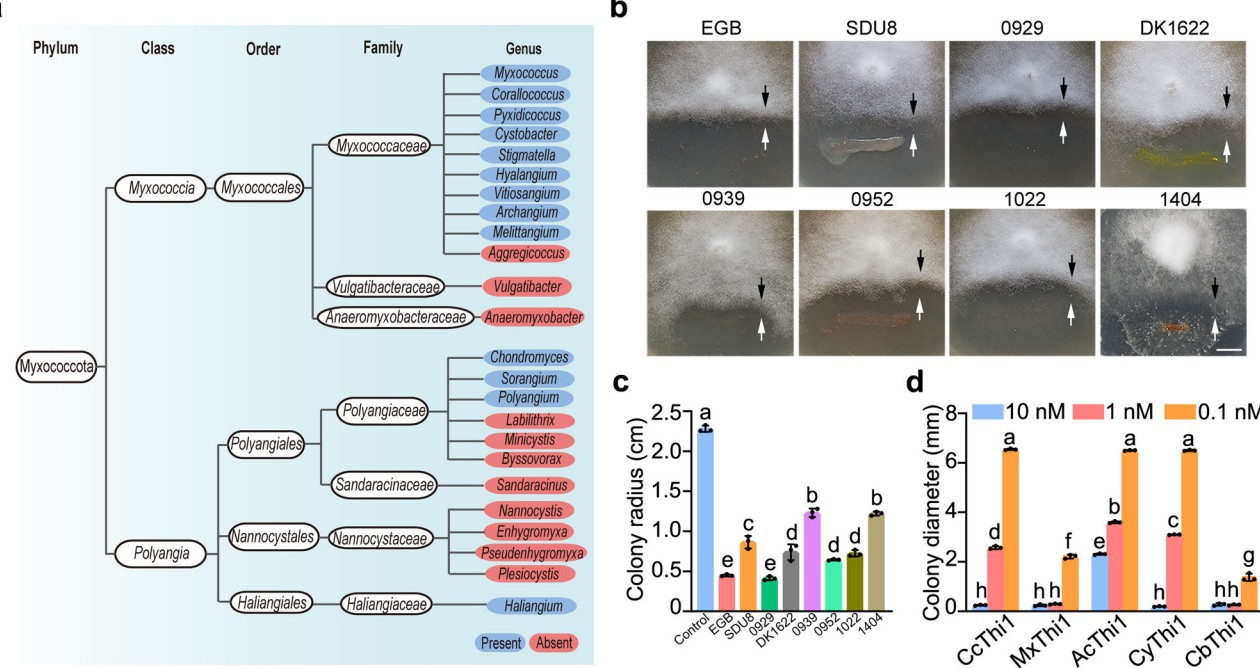

**Fig. 6 | Widely distributed homologs of thiaminase I CcThi1 in myxobacterial taxa are essential for interactions with *Phytophthora*. a** Distribution of CcThi1 homologs in myxobacterial taxa. The phylum Myxococcota currently contains 2 classes, 4 orders, 7 families, and 24 genera. Genera that harbor or lack CcThi1 homologs are indicated. **b** Co-culture assay of myxobacterial isolates with P6497 on VY/4 solid plates containing 10% V8 medium. Black arrows indicate the location of P6497, and white arrows indicate the location of myxobacteria. Scale bar: 5 mm. **c** Measurement of the colony radius of P6497 in the co-culture assay (**b**). The data represent the means ± SEM (*n* = 3 biological independent replicates). **d** Effects of CcThi1 homologs from DK1622 (MxThi1), *Archangium* sp. SDU8 (AcThi1) and

*Cystobacter* sp. 1404 (CyThi1) on the growth of P6497. The colony diameter of P6497 was measured after incubation at 25 °C for 20 h (Supplementary Fig. 17). Thiaminase CbThi1 from *Clostridium botulinum* was used as positive control. SDU8: *Archangium* sp. SDU8; 0929: *Myxococcus* sp. 0929; 0939: *Myxococcus* sp. 0939; 0952: *Myxococcus* sp. 0952; 1022: *Corallococcus* sp. 1022; 1404: *Cystobacter* sp. 1404. The data represent the means ± SEM (*n* = 3 biological independent replicates). Means within columns followed by different letters are significantly different in (**c**, **d**) (*p* < 0.05, one-way ANOVA, Duncan's multiple range test). Source data are provided as a Source Data file.

1.2 nmol/pot CcThi1 every other day decreased significantly by 91.44% compared to the control (Fig. 5d). Furthermore, obvious positive correlations were found between the thiamine concentration and disease index (*p* = 0.0005, $R^2$ = 0.543) or P6497 biomass (*p* = 0.0026, $R^2$ = 0.442), as well as between P6497 biomass and disease index (*p* < 0.0001, $R^2$ = 0.896) (Fig. 5e–g).

## CcThi1 and its homologs mediate the antagonistic interactions of myxobacteria and *Phytophthora*

After confirmation that thiaminase CcThi1 from strain EGB inhibits the mycelial growth of *Phytophthora* and controls root rot by thiamine regulation, we assessed the prevalence of CcThi1 homologs in the phylum Myxococcota by bioinformatic analysis of available genome sequences[24]. Bioinformatics analysis revealed that homologs of CcThi1 were widely distributed in 2 classes, including 3 orders, 3 families, and 13 genera (Fig. 6a). We selected 138 sequences to construct a phylogenetic tree (Supplementary Fig. 16a) and found 129 amino acid sequences with lipoprotein signal peptides. Among the 9 sequences without a lipoprotein signal peptide, 3 sequences without any signal peptide were found in the genera *Archangium* and *Sorangium*, while the remaining 6 sequences with a non-lipoprotein signal peptide were distributed in the genera *Corallococcus* and *Pyxidicoccus*.

We speculated that thiaminase I plays a crucial role in the antagonistic interactions between myxobacteria and *Phytophthora*. Seven myxobacterial species from the *Corallococcus*, *Archangium*, *Cystobacter*, and *Myxococcus* were selected and co-cultured with P6497, and all seven strains demonstrated significant antagonistic effects like strain EGB (Fig. 6b, c). To further verify their function, we heterologously expressed CcThi1 homologs from *M. xanthus* DK1622 (MxThi1),

*Archangium* sp. SDU8 (AcThi1), and *Cystobacter* sp. 1404 (CyThi1), as well as the unrelated Gram-positive *Clostridium botulinum* (CbThi1). All these recombinant proteins exhibited thiaminase I activity and antagonized the mycelial growth of P6497 at a concentration as low as 1 nM (Fig. 6d and Supplementary Figs. 17 and 18). These findings demonstrate that thiaminase I is widespread in myxobacterial taxa and is an essential factor in their antagonization against *Phytophthora*.

## Discussion

The presence of public goods in the plant-associated microbial community offers resources that are easily exploited by phytopathogens for their own benefit. Nevertheless, little is known about the ecological strategies adopted by the plant microbiome in response to this phenomenon. As keystone taxa in the ecosystem[29], myxobacteria employ a wealth of tactics to achieve ultimate victory in this battle. Here, we identified that the novel thiaminase I CcThi1 deployed by myxobacteria plays a critical role in their antagonistic interaction with *Phytophthora*. During this interaction, myxobacteria efficiently deplete the required thiamine and thus arrest the thiamine-sharing behavior of *P. sojae* from plant and soil microbes by employing thiaminase I. This thiaminase-mediated antagonistic interaction between myxobacteria and auxotrophy oomycete provides a profound understanding of the biological significance of public thiamine regulation in the natural environment.

Thiaminase I homologs are sporadically distributed in various prokaryotic and some eukaryotic organisms, especially in the phylum Myxococcota and orders Polyangiales, Haliangiales, Eubacteriales and Rotifera (Supplementary Fig. 16b). Thiaminase I from *Paenibacillus thiaminolyticus* (PDB: 2THI)[34] and *Clostridium botulinum* (PDB: 4KYS)[35]

belonging to Firmicutes were investigated in details, as well that from rotifers[45]. Notably, most myxobacterial thiaminases are lipoproteins, while those from other organisms are extracellular soluble proteins. This is especially notable since myxobacteria are Gram-negative organisms and have an outer membrane. Usually, Gram-negative bacteria do not need to translocate periplasmic proteins like CcThi1 to the outer membrane because proteins can be retained proximally to the cell by periplasmic localization. In the case of CcThi1, we hypothesize that the protein could be released to the environment by insertion into the outer membrane and budding. OMVs are related to cell killing in myxobacterial predation[43]. To achieve this function, enzymes, and secondary metabolites are encapsulated in OMVs[43,46]. We deduced that the translocation of myxobacterial thiaminase I through OMVs is an effective and protective pathway for the release of antagonistic factors by myxobacteria. Although CcThi1 inhibits *Phytophthora* growth and reduces the diameter of disease spots, direct contact between myxobacteria and *Phytophthora* is necessary to efficiently achieve maximal inhibition by secreting the thiaminase I CcThi1 via OMVs, as the high concentration of OMVs allows myxobacteria to interact efficiently with *Phytophthora* within the local position.

Evolutionary cluster analysis showed that reported thiaminases are divided into six groups with distinct geographical distributions in intestinal, aquatic, and soil ecosystems, whereby CcThi1 homologs from Myxococcota mainly belong to groups III-1 and III-2 (Supplementary Fig. 16b). In addition, obvious habitat-specific dispersion of thiaminase I producing organisms is observed, including mainly soil-dwelling myxobacteria, as well as aquatic cyanobacteria and rotifers, or intestinal Clostridiales[47,48]. Both CcThi1 from myxobacteria and CbThi1 from Clostridiales could decompose thiamine and inhibit the growth of *Phytophthora* (Fig. 6d), indicating their potential ecological functions. The biological function of thiaminase I has long been a mystery. Studies have shown that thiamine deficiency in Great Lakes salmonines, polioencephalomalacia in sheep, and cerebrocortical necrosis in cattle might be related to thiaminase I produced by intestinal bacteria[49–51]. Recently, thiaminase I in *Burkholderia thailandensis* was proposed to promote the salvaging of thiamine precursors from the environment[52]. However, TenA family thiaminases (type II) involved in the thiamine salvage pathway are also present in thiaminase I producers, such as *P. thiaminolyticus* (WP_119795742) and *Clostridium botulinum* (WP_019279372). Hence, the real biological functions of thiaminase I still warrant further investigation. Here, we found that CcThi1 participates in the antagonization of *Phytophthora* by depleting thiamine in rhizosphere soil, indicating its role in trophic interactions in the plant rhizosphere.

Trophic interactions play critical roles in shaping the microbiome[14]. It is widely considered that rhizodeposition regulates the plant microbiome, and organic amendments (OAs) have been shown to regulate the soil microbiome. However, an in-depth understanding of their effects on different trophic groups would be an important step toward the sound use of OAs[53]. Microbial interactions via metabolic competition or cooperation are commonly tested at the cellular or community levels in recent studies[7,22]. Fully understanding the physiological and biochemical basis of these regulation processes requires deeper insights at the community level. Thiamine is widely detected in aquatic and terrestrial environments, where it is utilized by thiamine auxotrophs[33]. Considering its low concentration in the environment, the thiamine-associated regulation of gene expression by riboswitches and similar mechanisms[54] might be negligible in the thiamine producers in the natural community, instead, by possible public thiamine regulation. By decomposing environmental thiamine, myxobacteria limit the availability of thiamine to *Phytophthora*. Hence, we present a working model for the control of *P. sojae* infection (Supplementary Fig. 19). Myxobacteria secrete CcThi1 into the soil embedded in OMVs, and it subsequently scavenges soil thiamine, and low levels of soil thiamine restrict the saprophytic growth of *P. sojae* and thereby inhibits the infection of soybean plants. The working principle of thiaminase I in this study represents an interspecific substrate-level regulatory mechanism for public goods in the soil microbiome. In the meantime, considering the migration characters of *Phytophthora* zoospores[12,13], thiaminase-mediated thiamine scavenging also provides fresh insight into the control of oomycete diseases.

Soil experiments showed that purified CcThi1 exhibits limited biocontrol efficiency compared to intact myxobacterial cells, although the thiamine decreased to a much lower level (Fig. 5b). However, thiamine content and *P. sojae* biomass from the rhizosphere soil showed that CcThi1-mediated thiamine scavenging is not the only factor regulating *P. sojae* population (Fig. 5d). In our previous work, we observed that the regulation of the soil microbiome driven by predation promotes the biocontrol of *Fusarium* wilt[28]. We deduced that the community adaption to myxobacterial treatments also contributes to the control of *Phytophthora* root rot in soybeans. The effect of thiamine as a public metabolite on the organization of the plant microbiome merits further investigation in the future. In our work, a small but significant difference in thiamine contents in root exudates of soybean cultivars (Supplementary Fig. 1d) was consistent with their resistance to *P. sojae*[55]. It may therefore be pertinent to investigate the thiamine secretion ability of soybean cultivars with different resistance levels to test the relationship between thiamine secretion and *P. sojae* resistance on a large scale.

## Methods

### Strains and culture conditions

*Corallococcus* sp. strains EGB and other myxobacterial isolates were cultured on VY/4 solid plates[25] or in liquid LBS medium (0.7% starch, 0.5% yeast extract, 0.1% tryptone, 0.1% MgSO₄) at 30 °C for 2 d. The model strain *Myxococcus xanthus* DK1622 and related transformants were cultured on CYE[25] plates at 28 °C. *P. sojae* wild-type P6497 and its GFP-labeled transformant were cultured on 10% V8 plates[56] at 25 °C. Liquid 10% V8 medium was used to collect the mycelia of *Phytophthora*; zoospores were prepared by repeatedly washing 3-day-old hyphae incubated in 10% V8 broth with sterile water and incubating the washed mycelia in the dark at 25 °C for 4 to 8 h until sporangia developed on most of the hyphae[57]. All strains, plasmids, and primers used in this study are listed in Supplementary Tables 1–4.

### Thiamine auxotrophy assay

The auxotrophy of *Phytophthora* was assayed using defined P1 medium[58] with slight modifications (0.2% casein hydrolyzate, Vitamins free; 1.5% sucrose; 0.01% MgSO₄·7H₂O; 0.0001% FeSO₄·7H₂O; 0.043% KH₂PO₄; 0.03% K₂HPO₄; 0.01% CaCl₂·2H₂O; 0.0001% ZnSO₄·7H₂O; 0.000002% CuSO₄·5H₂O, NaMoO₄·2H₂O, MnCl₂·4H₂O; pH 6.4). When necessary, 2 μM thiamine, thiamine monophosphate (TMP) and thiamine pyrophosphate (TPP), 1 μM hydroxymethylpyrimidine (HMP) and hydroxyethylthiazole (HET), 1 mg/mL root exudates and 5 mg/mL supernatant of thiamine-producing strain T-1 were added to the P1 medium for *Phytophthora* cultivation. The collection of root exudates and isolation of thiamine-producing bacteria are described in the Supplementary Material. For solid plate co-culture, strain T-1 (2 μL, 10⁶ cell/mL) was inoculated at a distance of 1.5 cm from the *Phytophthora* mycelial disks and incubated for 7 d. The biomass of *Phytophthora* was quantified by qPCR. All quantitative PCR amplifications were conducted on the Applied Biosystems 7500 Real-Time PCR system using AceQ Universal SYBR qPCR Master Mix (Vazyme, China). For culture of thiamine auxotrophic *E. coli* K-12 Δ*thiE* and thiamine auxotrophic *Saccharomyces cerevisiae* BY4743 Δ*thi6*, we first pretreated the MM or Delft-1 medium (0.01 or 1 μM thiamine addition) with 2 μM Ccthi1 at 30 °C for 4 h, following by incubation of K-12 Δ*thiE* and BY4743 Δ*thi6* with a final concentration of 10⁶ cells/mL and 10⁵ cells/mL, respectively. Plates were analyzed after 20 h of incubation. Cell viability was tested via serial dilution followed by counting

bacterial and fungal colonies after incubation on LB or YPD agar plates. Medium without thiamine addition was used as a control.

## Myxobacteria–*Phytophthora* confrontation assays

Co-cultures of *Phytophthora* with different myxobacteria were carried out on a solid VY/4 medium supplemented with V8 when necessary. In this co-culture system on solid media, the mycelial block ($0.2 × 0.2$ cm) was removed from the growing edge of the *Phytophthora* colony, and myxobacterial strains (5 µL, $10^7$ cell/mL) were inoculated at a distance of 2 cm from the mycelia block. The colony diameter was measured after incubation at 25 °C for 7 d. Additionally, membrane-separated co-incubation of the strains EGB and P6497 or GFP-labeled was carried out to test if direct cell contact is required for the inhibition effect (molecular cut-off 7 kDa or 0.22 µm)[25]. The observation of GFP-labeled P6497 was taken by a stereoscopic fluorescence microscope (Nikon, SMZ25, Japan).

For co-cultural assays in liquid media, the freshly prepared P6497 mycelia (0.1 g, wet weight) was transferred into TV medium (15 mL TPM buffer[25], 5 mL LBS, 5 mL 10% V8) containing strain EGB with a final concentration of $10^5$ cells/mL, followed by culture at 30 °C and 180 rpm for 3 d. The biomass of strain EGB and P6497 was quantified by qPCR. The mycelia after 3 d of co-culture was visualized by scanning electron microscopy (SEM, Hitachi SU8010).

To identify the effects of strain EGB on the zoospores production of P6497, the prepared P6497 mycelia (50 mg, wet weight) was co-cultured with strain EGB ($10^5$ cell/mL) in 50 mL TV medium at 30 °C and 180 rpm for 1 h. The zoospores were prepared by repeatedly washing with sterile water as described above[57], and the numbers of the zoospores were counted directly via light microscopy.

## Cell death assays

For the preparation of the cell-free supernatant (SUP), strain EGB was cultured in LBS medium at 30 °C for 2 d, SUP was collected by centrifugation at $12,000g$ for 30 min, followed by freeze drying for preparation of the concentrated SUP that was re-dissolved in sterile water (SUPL). To determine the effects of SUPL on the activity of *P. sojae* mycelia, GFP-labeled P6497 and P6497 (0.1 g, wet weight) were cultured with SUPL (10 mg/mL, 1 mL) at 25 °C. After 1 d incubation, the fluorescence detection was performed using confocal laser scanning microscopy (CLSM, LeicaTCSSP3, Germany) with an excitation wavelength of 488 nm and emission wavelength of 507 nm. In addition, the cell viability of P6497 was assessed using the MTT assay. The freshly prepared P6497 mycelia was mixed with SUPL (10 mg/mL) or sterile water at 25 °C for 24 h, followed by washing 3 times with sterile water to ensure that SUPL was completely removed from the mycelia. Then, 1 mL MTT (3-(4,5-dimethylthiazol-2-yl)-2,5-diphenyltetrazolium bromide, 0.5 mg/mL) was added to the mycelia and allowed to react at room temperature for 90 min[59]. Free MTT was subsequently removed with sterile water, and the mycelia were incubated with 1 mL absolute ethanol at room temperature overnight. The absorbance of the color reaction at 570 nm ($OD_{570}$) was measured using a SpectraMax i3x spectrophotometer (Molecular Devices, Austria). After 3 d incubation, detailed observation of the mycelia was realized by scanning electron microscopy (SEM, Hitachi SU8010).

## Purification of the CcThi1 thiaminase from strain EGB

Strain EGB was cultured in the LBS medium described above, after which the cells were harvested by centrifugation (10000 g for 10 min) and washed with deionized water to remove the residual medium. The cells were re-suspended in 50 mM Tris-HCl buffer (pH 7.2) and disrupted by ultrasonication (Sonicator 201 M, Kubota, Japan) at 4 °C. The crude lysate was cleared by centrifugation at $12,000g$ for 20 min, followed by ultracentrifugation ($150,000g$) at 4 °C for 2 h[60]. The collected precipitate from ultracentrifugation was the membrane fraction, which was re-suspended in Buffer A (50 mM Tris-HCl pH 7.4; 100 mM NaCl; 1% CHAPS) to obtain the solution, followed by ultracentrifugation at $150,000g$ at

4 °C for 2 h to remove insoluble compounds. The obtained protein fraction was dialyzed against Buffer B (50 mM Tris-HCl pH 7.6; 0.6% CHAPS) and fractionated by ion exchange chromatography (IEC) on a Capto Q column (GE Healthcare), eluted with a linear sodium chloride gradient using Buffers B and C (50 mM Tris-HCl pH 7.6; 0.6% CHAPS; 1 M NaCl, pH 7.6) at a flow rate of 2 mL/min. Fractions with anti-*Phytophthora* activity were collected. Selected fractions were further collected and dialyzed against Buffer D (50 mM Tris-HCl pH 7.2; 0.6% CHAPS; 100 mM NaCl) and subjected to hydroxyapatite chromatography (HC) on a Bio-Scale™ mini CHT™ column (Bio-Rad) with Buffers D and E (50 mM Tris-HCl; 0.6% CHAPS; 100 mM NaCl; 20 mM phosphate buffer, pH 7.2) at a flow rate of 0.2 mL/min. The active fractions were concentrated using a 10 kDa molecular mass centrifugal filter (Centriprep®, Millipore) and further purified by gel filtration chromatography (GFC) on a HiLoad 16/600 Superdex 200 pg column (GE Healthcare) with Buffer D at a flow rate of 1 mL/min. The active fractions with anti-*Phytophthora* activity were analyzed by SDS-PAGE, and the stained gel band corresponding to the results of the growth inhibition assay with P6497 was excised and analyzed by NanoLC–ESI–MS/MS[61]. Fractions with anti-*Phytophthora* activity were analyzed in 96-well assays.

## Expression and purification of recombinant thiaminase I

The gene encoding CcThi1 and its homologs were amplified by PCR and ligated by homologous recombination into pET29a (+) between the *Eco*R I and *Hin*d III restriction sites. For CcThi1$^{CI48A;E269A}$, site-directed mutagenesis was applied using homologous recombination by amplification of the entire circular plasmid[62] (Mut Express II Fast Mutagenesis Kit V2, Vazyme). The primers used for amplification are shown in Table S3. *E. coli* BL21 (DE3) was transformed with the constructed vectors to produce the recombination strain, and the expression of recombinant proteins in *E. coli* was induced by adding isopropyl β-D-thiogalactoside (IPTG) to a final concertation 0.2 mM at 16 °C. Cells were collected and sonicated in 50 mM Tris-HCl buffer (pH = 7.2) after 20 h of induction. The cell lysate was centrifuged at $12,000g$ for 20 min at 4 °C. Purification of recombinant protein from the cleared cell lysate was performed by affinity chromatography using a Ni Sepharose 6 Fast Flow column (GE Healthcare), which was eluted with 200 mM imidazole. The protein was dialyzed against 50 mM Tris-HCl buffer (pH=7.2) before use for viability assays.

## Growth inhibition assay

The effects of SUP, SUPL, purified fractions, and recombinant proteins on the growth of P6497 were evaluated at different concentrations. A mycelial block from a fresh colony of P6497 or the prepared zoospores was cultured on a 10% V8 agar medium supplied with recombinant proteins, followed by incubation at 25 °C for 12 or 7 d. Then, the colony diameter and the length of germ tubes were recorded. To determine the effects of thiamine supplement on the growth of zoospores and P6497 mycelia with presence of CcThi1, prepared zoospores was cultured on 10% V8 agar medium supplied with thiamine (0, $2.96 × 10^{-3}$, $2.96 × 10^{-2}$, $2.96 × 10^{-1}$ and 2.96 µM) and 0.24 nM CcThi1, followed by incubation at 25 °C for 7 or 12 d, and the length of germ tubes or colony diameter were recorded as described above. For the growth inhibition assay, agar discs ($0.2 × 0.2$ cm) were removed from the edge of actively growing *P. sojae* colonies and transferred to the center of 96-well plates containing 90 µL liquid V8 medium and 10 µL SUPL or other enzymes in each well at different concentrations. The growing *P. sojae* mycelia were observed using a stereomicroscope (Nikon SMZ-10) after 20 h of incubation at 25 °C, and the colony diameter of *Phytophthora* in each well was measured.

## Thiaminase I activity assay and determination of the thiamine content

Thiaminase I activity was determined as described previously[63]. A reaction mixture comprising 194 µL of TCEP buffer (100 mM NaCl, 50 mM

phosphate buffer (pH 7.2), 10 mM TCEP, 200 μM 4-nitrothiophenol (4-NTP), 400 μM thiamine) and 6 μL of the purified protein solution was incubated at 37 °C, and $OD_{411}$ value of the reaction was measured each minute. An otherwise identical reaction solution without thiamine was included as a control. 4-NTP can replace the thiazole base as the nucleophile in the enzymatic reaction of thiaminase I, and the consumption rate of 4-NTP is correlated with thiaminase I activity. To determine the kinetic parameters, 1–100 μM of the thiamine was incubated with CcThi1 or SP-CcThi1 under optimal conditions. The $K_m$ and $V_{max}$ values were calculated from the Michaelis-Menten equation analysis using GraphPad Prism 8.0.2 (GraphPad Software Inc., USA).

To identify the products of enzymatic reactions in the absence of a nucleophile, a mixture of CcThi1 (1.19 μM) and thiamine (0.29 mM) was incubated at 25 °C for 6 h, followed by heating at 95 °C for 5 min to inactivate the enzyme. The enzymolysis products were then analyzed by UPLC-MS (described in the Supplementary Material).

For determination of the thiamine content, fluorometric and bioassay methods were conducted as described before[64,65]. The intact thiamine was oxidized to fluorescent thiochrome in the presence of potassium ferrocyanide and sodium hydroxide. Briefly, a 5 mL sample was combined with 3 mL of 15% NaOH or basic potassium ferricyanide solution and shaken for 15 s. Then, 10 mL of n-butanol was added, shaken vigorously for 90 s, and rested for 5 min to allow stratification. The fluorescence of the upper organic phase was measured using a microplate reader (SpectraMax i3x) with an excitation wavelength of 365 nm and an emission wavelength of 435 nm. A sample comprising 0.29 μM thiamine was included as a reference standard. *E. coli* K-12 Δ*thiE* deficient in thiamine synthesis was used as a reference strain, as its growth was consistent with the supplied thiamine content in the bioassay. Hence, the $OD_{600}$ value of the K-12 Δ*thiE* culture was utilized to indirectly determine the thiamine concentration. *E. coli* K-12 Δ*thiE* was cultured in liquid 48-well plates with 250 μL of minimal medium (MM) and 250 μL of the thiamine complementation solution (supernatant of strain T-1, root exudates, zoospores lysate, soil water extract, or standard solution with a concentration of 0–100 nM) at 37 °C and 180 rpm for 8 h, and the biomass of *E. coli* K-12 was determined by measuring the $OD_{600}$.

## Bio-control assay

Hypocotyl inoculation was used to test the biocontrol abilities of strain EGB in pot experiments under greenhouse conditions (16 h-light/8 h-dark, 80% relative humidity) with two soybean cultivars (Hefeng-47 and Taiwan-292)[66]. Soybeans were grown in coarse vermiculite until the cotyledons expanded (about 6 d). Then, a small wound about 1 cm long was produced in the hypocotyl and inoculated with a 4-mm-diameter agar disc of *P. sojae*. The EGB suspension ($10^8$ cells/mL, 5 mL) or sterile water was supplied to the roots along the hypocotyl. Three weeks after planting, disease severity in seedlings (stem rot), including the control efficiency and disease index of Hefeng-47 were analyzed according to a five-point scoring system: 1 = 1–10%, 2 = 11–35%, 3 = 36–65%, 4 = 66–90%, and 5 = 91–100% of the stem necrotic or browning area. For Taiwan-292, we used shoot height as an evaluation criterion due to its lodging resistance.

To further determine the biocontrol activity of myxobacteria and thiaminase CcThi1 against *P. sojae* in the soil environment, we used the inoculum layer method[44] to evaluate the ecological effect of CcThi1 and myxobacteria. Soybean seeds (Hefeng-47 cultivar) were placed in 200*g* of moist soil and incubated at 25 °C with 80% relative humidity under a 16 h-light/8h-dark photoperiod, and soybean seedlings with uniform growth were selected for the following pot experiments. *P. sojae* was cultivated on 10% V8 medium until the colony covered the surface (about 10 d, 9 cm colony diameter). Then, the intact agar culture was placed on the vermiculite surface and covered with 5 cm of coarse vermiculite. Two-week-old soybean seedlings were transferred to pots containing 10% V8 plates or *P. sojae* plates. In parallel, the roots

were inoculated with 5 mL of myxobacterial suspension ($10^8$ cells/mL) of strains EGB, DK1622, CL1003, and CL1004 or irrigated with CcThi1 (1.2 nmol/pot every other day). The incidence of lesions was photographed and recorded after 21 d of transplanting. The following rating system was used to assess the soybean root rot disease incidence as modified from Thomas et al.[44]: 1, no root rot; 2, 10–20% root rot; 3, 20–40% root rot; 4, 40–60% root rot; 5, 60–80% root rot; 6, 80–100% root rot area or seedling dead. The calculation of disease index and control efficiency was reported previously[25].

Soil was collected around the roots of the soybean plants for thiamine content determination every 7 d after transplanting. A sample comprising 10 g of soil was accurately weighed, suspended in 10 mL of deionized water, and agitated at 200 rpm overnight. The supernatant was collected by centrifugation for the thiamine bioassay, as described above. At the same time, the abundance of *P. sojae* in the collected soil was analyzed by qPCR. Briefly, DNA was extracted from the soil samples (0.5 g) using a soil DNA extraction kit (FastDNA SPIN Kit, MP Biomedicals, USA), and *P. sojae* biomass was quantified by qPCR using the specific primers listed in Supplementary Table 3. Soil samples were collected on day 21 after transplanting.

In vitro leaf infection experiment, uniformly sized leaves from 10-day-old soybean plants were chosen. Then, 20 μL cell suspension of strain EGB ($10^8$ cells/mL), or 50 μL thiaminase I CcThi1 (0.01, 0.15, and 0.29 μM) were placed on the leaf surface, followed by covering with P6497 agar blocks (0.2 cm × 0.2 cm) or suppling with zoospores with a final concentration of $2 \times 10^4$ cells. 20 μL $H_2O$ covered with P6497 agar blocks and 50 μL buffer (Tris-HCl, 7.2, 20 mM) supplied with $2 \times 10^4$ zoospores were used as control. All the treatments of P6497 agar blocks and zoospores were supplied with 200 μM thiamine, with the aim of identifying the effects of thiamine addition on the P6497 infection. All the leaves were cultivated at 25 °C with 80% relative humidity under a 16 h-light/8 h-dark photoperiod. After 36 h of incubation, the lesion length of the leaves from different treatments was determined and then stained with lactophenol trypan blue (10 mL phenol, 10 mL glycerol, 10 mL lactic acid, 60 mL ethanol, and 40 mg of Taipan blue dye) at room temperature for 3 h. All leaves were decolorized overnight using chloral hydrate.

## Isolation of OMVs

Isolation of OMVs from EGB and *Myxococcus* strains was performed as reported previously[25]. Briefly, cells grown exponentially in liquid LBS (for EGB) or CTT (for *Myxococcus*) were obtained. To isolate cell-free supernatants, cells were harvested at 4 °C (14,000×*g* for 10 min), and the supernatant was filtered through a 0.22 μm sterile filter (termed SUP) and centrifuged at 150,000*g* at 4 °C for 2 h to pellet OMV. The resulting cell-free supernatants were regarded as OMV-free SUP (SUPU). The prepared SUP, SUPU, and various concentrations of OMVs were used for the determination of the thiaminase activity and anti-*Phytophthora* ability in the 96-well-plate bioassay. Since the thiaminase activity of SUP was extremely low, the reaction system was performed with 200 μL components and 200 μL reaction buffer at 37 °C for 1 h. The OMVs were then precipitated with 20% of ice-cold trichloroacetic acid (TCA) for 30 min, followed by 20 min centrifugation (14,000×*g* at 4 °C) and three wash steps with ice-cold acetone with centrifugation steps between washes (14,000×*g*, 5 min at 4 °C). The pellets were air-dried for 30 min and re-suspended in SDS loading buffer to a density equivalent to their respective cell samples and analyzed by SDS-PAGE. The thiaminase content in the membrane fraction and OMVs was further confirmed by western blotting with an anti-CcThi1 antibody[25]. The details of vector construction and western blot analysis are provided in the Supplementary Methods.

## Bioinformatic analysis

Phylogenetic tree analysis of related sequences was performed using MEGA version 7.0.26 with the maximum likelihood or neighbor-joining

method. Signal peptide prediction was performed using the SignalP 5.0 server. BLASTP searches were performed to identify homologs of CcThi1 across the phylum Myxococcota with *e*-values less than 1e-6 and query coverage values between 30 and 100% in the non-redundant NCBI protein database. To explore the distribution of thiaminase I homologs in the ecosystem, we ran a BLAST analysis of the genomes of different species using the *ccthi1* gene from strain EGB as the query. The screening criteria included an *e*-value of less than 1e−6 and coverage between 43 and 100%. ClustalW version 2.0 was used to identify the critical amino acid residues, and the resulting alignment was used to generate a WebLogo.

## Statistical analysis and reproducibility

All experiments were repeated independently at least three times where similar results were found. The data represent the means ± SEM of all biological replicates. The significance of differences ($p < 0.05$) was assessed using a two-sided Student's *t*-test or one-way analysis of variance (ANOVA) with Duncan's multiple range test in SPSS (ver. 21.0; IBM Corp., Armonk, NY, USA). Correlation analysis was performed using Pearson's *r* correlation in GraphPad Prism 8.0.2 (GraphPad Software Inc., USA). All gels, blots, and micrographs were repeated at least three times independently and showed similar results.

## Reporting summary

Further information on research design is available in the Nature Portfolio Reporting Summary linked to this article.

## Data availability

Raw sequencing data have been deposited in the NCBI SRA database (BioProject accession number: PRJNA909170). The nucleotide sequences of the *ccthi1* gene of strain EGB (Genbank: WP_223633232.1), the *mxthi1* gene of strain DK1622 (GenBank: ABF86831.1), the *acthi1* gene of *Archangium* sp. SDU8 (GenBank: QRO01607.1) and the *cythi1* gene of *Cystobacter* sp. 1404 (GenBank: OP978502) have been deposited in the NCBI nucleotide database. The data that support this study are available within the article and its Supplementary Information files. Source data are provided with this paper.

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

## Acknowledgements

We thank Professor Yuanchan Wang and Suomeng Dong (Nanjing Agricultural University) for generously providing *Phytophthora* strains. This work was supported by the National Natural Science Foundation of China (32170123 to Z.C., 32070027 to Z.L. and 32270066 to Z.L.), the Key Research & Development Plan of Jiangsu Province (BE2020340 to Z.C.), the Fundamental Research Funds for the Central Universities (YDZX2023014 and KYZZ2022001 to Z.L.), the National Science and Technology Major Project (2020ZX08009-04B to Z.C.), the National Key Research and Development Program of China (2021YFC2103600 to Z.C. and Z.L.) We also thank Mr. Gang Hu and Chun Qin (Electron Microscope Experimental Center, College of Life Sciences, Nanjing Agricultural University) for their help with SEM.

## Author contributions

C. Xia., Z.L., and Z.C. designed the experiments. C. Xia. performed the majority of the experiments. Y.Z., X.L., D.W., C.X., and Y.C. performed pot experiments. M.Q., X.Y., and X.G. isolated bacterial strains. L.Z. and J.W. performed gene knockout experiments. Y.H., D.D., H.C., and D.S. collected and analyzed the data. C. Xia., Z.L., and Z.C. wrote and revised the paper.

## Competing interests

The authors declare no competing interests.
