## [Peer Review File · Nature Communications]

Reviewers' Comments:

Reviewer #2:

Remarks to the Author:

The authors make a strong case for the importance of thiamine acquisition in the rhizosphere. The specific role of the EEB strain in inhibiting oomycete infection by limiting the availability of thiamine has been demonstrated in several experiments which are carefully described. The method of thiamin delivery to the environment is especially interesting. The effectiveness of this bacteria as in limiting disease development by oomycete pathogens, suggests that it might have a role as a biocontrol agent in either greenhouse or field applications. But the focus of this manuscript is rightly on the biological significance of competition for thiamine resources, since this is a topic that will be of interest to the largest reading audience. Auxotrophy for thiamine is a well-known property of phytophthora pathogens (Erwin and Ribeiro 1996), but this is the first manuscript that I am aware of that considers its biological significance. In fact, the enormity of the evidence provided by the authors forced me to think how oomycete soil pathogens have adapted to this selection pressure, a research direction not covered in this manuscript.

The most important difference between fungal pathogens and oomycetes is that swimming zoospores of oomycetes travel at speeds of 120-150 $\mu\text{m}/\text{sec}$ and can migrate through soil water or on the surface to find new hosts (Erwin and Ribeiro 1995). Newly released zoospores can swim for up to 24 h before encysting. In the absence of appropriate signals, a second zoospore can emerge from the cyst to continue swimming. Orientation to host roots can come from host specific signals or stress compounds such as the release of ethanol from roots under flooded conditions.

Early research suggested that swimming zoospores ran on existing stores and isotope tracing experiments indicated that they did not express transporters capable of acquiring sugars or amino acids. However swimming zoospores do uptake polyamines, and polyamines are excreted from plant roots (Chibucos and Morris 2006). Polyamines however do NOT function as chemotactic molecules. I think that this observation is significant for this work because in plants thiamine transporters are also polyamine transporters (Martinus et al 2016). Related transporter to PUTs are also found in Oomycetes. Polyamines like thiamine are also required for rapidly dividing cells. Zoospores that are generated and released from hyphae in a thiamine-poor environment have the opportunity to acquire additional thiamine resources in transit. The authors have also noted that thiamine degradation is widespread across bacteria (Supplementary Fig. 3C). Thus selection pressure to respond to thiamine depletion in the soil has existed for perhaps millions of years. It seems likely to me that swimming zoospores are also capable of scavenging thiamine while swimming, and if this can be demonstrated, this is an important and relatively simple adaptive strategy of oomycetes in response to this selection pressure, that fungal pathogens don't have. A second reason for asking the authors to explore this avenue of research is that the ability of Phytophthora zoospores to migrate in the soil may well limit the effectiveness of a biocontrol strategy employing thiamine-degrading microbes in a field soil, but not as a foliar application to leaves or in greenhouse pots. I believe that this can be convincingly demonstrated by the authors and contribute to strengthening the value of this manuscript.

Suggested Experiments.

The highest concentration of zoospores of *Phytophthora sojae* are produced by transferring several agar plugs to a fresh V8 agar plate. Repeated washing of the plate (every 20 min) is initiated two-three days later when most of the plate surface has almost covered the agar surface. Thiamine content of zoospores can be manipulated as follows.

EGB colonization of hyphae producing sporangia may limit the production of zoosporangia or the thiamine stores delivered to zoospores. Here I suggesting to test what happens when EBG

In contrast does thiamine supplementation of zoospores improve their infectivity on soybean leaves (Supplementary fig 3)?

Does thiamine supplementation diminish the effectiveness of EGB as a foliar biocontrol agent ?

Overall impression

A significant omission of the abstract was that there was no mention that the thiaminase was excreted in vesicles.

In the introduction especially, I found the phrasing of ideas to be awkward,

I am not happy with the present title but with revisions, I still feel that this manuscript is a very good fit for Nature Communications

Other comments :

Lines 180-183 the results are interpreted as "leading to the death of P sojae.

Lines 340-341 Later they point out that the mechanism is not killing but inhibition of growth and also Fig. 2d

I found the images of Fig.2a difficult to interpret. It would be better if the legend included a sentence of explanation as I don't know what I am supposed to be seeing.

Lines 283-284...recombinant proteins against strain P6497, indicating that lipoation and membrane association are not necessary for its growth-inhibitory effect against Phytophthora

True but the efficiency of inhibition by direct contact supplementary 3F was much much higher and should have been memorably mentioned in the discussion.

Lines 291 -294 germination of zoospores (Supplementary Fig. 3e). Furthermore, CcThi1 effectively limited the infection of soybean leaves by strain P6497 in the biocontrol assay. The diameter of disease spots was significantly reduced with the increase of CcThi1 concentration (Supplementary Fig. 3f).

So is this negated by thiamin supplementation of swimming zoospores?

Entirely by accident I found possibly the earliest mention of the possible importance of thiamine to phytophthora ERWIN DC, KATZNELSON H. Suppression and stimulation of mycelial growth of Phytophthora cryptogea by certain thiamine-requiring and thiamine-synthesizing bacteria. Can J Microbiol. 1961 Dec;7:945-50. doi: 10.1139/m61-119. PMID: 13890710. I only knew of the thiamin requirement from the text cited below.

My interest in zoospores within the oomycete research community is well known. Combining that with the references that I felt needed to be cited is a clear indication of my identity.

References

Chibucos MC, Morris PF. Levels of polyamines and kinetic characterization of their uptake in the soybean pathogen *Phytophthora sojae*. Appl Environ Microbiol. 2006 May;72(5):3350-6. doi: 10.1128/AEM.72.5.3350-3356.2006. PMID: 16672477; PMCID: PMC1472313.

D. C. Erwin and O. K. Ribeiro. 1996. *Phytophthora Diseases Worldwide* 592pp. St The American Phytopathological Society, Paul, MN, USA:

Martinis J, Gas-Pascual E, Szydlowski N, Crèvecoeur M, Gisler A, Bürkle L, Fitzpatrick TB. Long-Distance Transport of Thiamine (Vitamin B1) Is Concomitant with That of Polyamines. Plant Physiol. 2016 May;171(1):542-53. doi: 10.1104/pp.16.00009. Epub 2016 Mar 22. PMID: 27006489; PMCID: PMC4854701.

Mulangi V, Chibucos MC, Phuntumart V, Morris PF. Kinetic and phylogenetic analysis of plant polyamine uptake transporters. Planta. 2012 Oct;236(4):1261-73. doi: 10.1007/s00425-012-1668-0. Epub 2012 Jun 19. PMID: 22711282.

Reviewer #3:

Remarks to the Author:

This impressive manuscript describes an extensive investigation into the importance of thiamine for oomycete disease of soybean, and how thiaminases secreted by myxobacteria can protect against disease by inhibiting oomycete growth.

This is a very important piece of work for the field. It is very thorough and provides a wealth of data that deliver a mechanistic understanding of interactions between bacteria, oomycetes and plants mediated by a specific metabolite. The findings are broadly relevant to soil ecosystems, with potential applications in biocontrol and sustainable agriculture.

The manuscript is generally well-written, but would benefit from English language editing as many sentences had minor grammatical problems.

The taxonomy of the myxobacteria as presented in line 86 and around line 513 is wrong. Myxobacteria were recently reclassified as a distinct phylum from Deltaproteobacteria called the

Myxococcota - see Waite et al (2020) and Oren and Garrity (2021), which also affected the number of myxobacterial suborders.

I found the introduction a bit confusing. It read like a list of statements about microbiomes and there wasn't a clear logic running through it. Does the microbiome dictate which metabolites are found in the public commons, or vice-versa? If public goods drive microbiome resilience, how does mono-cropping render microbiome structure unstable, and why is the accumulation of pathogens an indicator of instability?

I was also confused by the presentation of the earliest results sections, which seemed unnecessarily complicated. Why use a bioassay for thiamine, when later it is detected directly by UPLC-MS? Why use strain T-1 to put thiamine into the system, when pure thiamine could have been added directly? When using T-1 (Fig 1c), a better control than no *E. coli*, would have been an *E. coli* which doesn't secrete thiamine. In Fig 1d, why is there as much growth with water for two of the species as there is with thiamine - only one of the species seems to respond as expected to the water negative control. In Fig 1b, it doesn't look like *P. sojae* is being stimulated by any of the additives - it seems hard to reconcile the results in Fig 1b with the data in Fig 1d. In Fig 1c - how is the biomass of phytophthora distinguished from that of *E. coli*?

I think it is difficult to claim that EGB's effect on *P. sojae* is not predatory, as the co-incubation experiments only seem to last 24h. I assume that predation is dismissed as there is no significant growth of EGB, but 24h is not a long time for myxobacteria to start showing signs of active predation - in our experience many predatory strains take 2-3 days to start exhibiting significant growth when preying upon susceptible organisms.

Can the addition of thiamine stop the inhibitory effect of adding EGB? Line 203: Without showing that at this point in the paper it is difficult to conclude that it is specifically thiamine and not some other secretion of EGB that is inhibiting oomycete growth, which makes the conclusion in line 213-214 difficult to justify.

What is the significance of thiaminase being secreted in OMVs? If it is periplasmic in the cell, would it not be in the lumen of the OMV? How could it access thiamine substrate? Is the OMV permeable to thiamine, or do the OMVs lyse to release the enzyme?

Are myxobacteria generally prototrophic for thiamine?

Minor comments

Line 91 - in what way is it novel?

Line 92 - it is not strictly increasing the plant's resistance to Phytophthora, but reducing the amount of pathogen.

Line 95 - The statement about HGT seems throwaway. What do you mean by this statement? How is it relevant?

Line 96-97. In what way is it unique? And what do you mean by fine-tuned? Fine-tuned implies that the concentration of thiamine and thiaminase in soil are tightly-regulated, but that hasn't been shown - rather that small changes in thiamine can have substantial impacts on oomycete growth and pathogenesis.

Line 172 - words like 'plunder' and 'greedily' seem excessively poetic/emotive. (also 'pivotal' line 235, 'loopholes' line 557 and 'raiding' - several places including abstract).

Line 178 - why would suppression of *P. sojae* growth be unexpected when EGB is a known predator?

Fig 2 - I found it difficult to relate panel b to panel a. Panel c didn't seem to add much to the manuscript. In panel d, rather than comparing with/without EGB extract, it would have been more interesting to see what the mycelia looked like at the edge of where growth was being inhibited by EGB in the experiments of panel a.

Line 280. What do you mean by 'functional confirmation'? At this point in the manuscript the gene's function has not been confirmed.

Line 300. It would be more informative to present Km values rather than specific activities, as Km can be related to ecologically relevant concentrations of substrate.

Line 491 - the authors state that thiaminase is predominantly (86%) in OMVs, but the western

blot in Fig 4 seems to suggest otherwise.

Line 610-612. I didn't understand this sentence or it's significance.

Response to review's suggestions:

Reviewer #2 (Remarks to the Author):

The authors make a strong case for the importance of thiamine acquisition in the
rhizosphere. The specific role of the EEB strain in inhibiting oomycete infection by limiting
the availability of thiamine has been demonstrated in several experiments which are
carefully described. The method of thiamin delivery to the environment is especially
interesting. The effectiveness of this bacteria as in limiting disease development by
oomycete pathogens, suggests that it might have a role as a biocontrol agent in either
greenhouse or field applications. But the focus of this manuscript is rightly on the biological
significance of competition for thiamine resources, since this is a topic that will be of interest
to the largest reading audience. Auxotrophy for thiamine is a well-known property of
phytophthora pathogens (Erwin and Ribeiro 1996), but this is the first manuscript that I am
aware of that considers it biological significance. In fact, the enormity of the evidence
provided by the authors forced me to think how oomycete soil pathogens have adapted to
this selection pressure, a research direction not covered in this manuscript.

The most important difference between fungal pathogens and oomycetes is that swimming
zoospores of oomycetes travel at speeds of 120-150 $\mu\text{m}/\text{sec}$ and can migrate through soil
water or on the surface to find new hosts (Erwin and Ribeiro 1995). Newly released
zoospores can swim for up to 24 h before encysting. In the absence of appropriate signals,
a second zoospore can emerge from the cyst to continue swimming. Orientation to host
roots can come from host specific signals or stress compounds such as the release of
ethanol from roots under flooded conditions.

Early research suggested that swimming zoospores ran on existing stores and isotope
tracing experiments indicated that they did not express transporters capable of acquiring
sugars or amino acids. However swimming zoospores do uptake polyamines, and
polyamines are excreted from plant roots (Chibucos and Morris 2006). Polyamines
however do NOT function as chemotactic molecules. I think that this observation is
significant for this work because in plants thiamine transporters are also polyamine
transporters (Martinus et al 2016). Related transporter to PUTs are also found in
Oomycetes. Polyamines like thiamine are also required for rapidly dividing cells. Zoospores
that are generated and released from hyphae in a thiamine-poor environment have the
opportunity to acquire additional thiamine resources in transit. The authors have also noted

that thiamine degradation is widespread across bacteria (Supplementary Fig. 3C). Thus
selection pressure to respond to thiamine depletion in the soil has existed for perhaps
millions of years.

**Response:** Thank you for your recognition of the scientific significance of our
work. Adaptation of soil oomycetes pathogens to thiamine depletion pressure is a
valuable issue for the adaptive evolution. This should inspire the interest of
evolutionary biologists. Further experiments will be designed in our following research
to investigate this issue.

It seems likely to me that swimming zoospores are also capable of scavenging thiamine
while swimming, and if this can be demonstrated, this is an important and relatively simple
adaptive strategy of oomycetes in response to this selection pressure, that fungal
pathogens don't have. A second reason for asking the authors to explore this avenue of
research is that the ability of *Phytophthora* zoospores to migrate in the soil may well limit
the effectiveness of a biocontrol strategy employing thiamine-degrading microbes in a field
soil, but not as a foliar application to leaves or in greenhouse pots. I believe that this can
be convincingly demonstrated by the authors and contribute to strengthening the value of
this manuscript.

**Response:** Although *Phytophthora* zoospores might escape thiamine limitation by
migration, thiaminase I could diffuse into the environment to decrease thiamine
availability. The key is the population of myxobacteria in the rhizosphere and the
expression and secretion levels of thiaminase I. We will set field tests to evaluate the
effectiveness of the thiamine scavenging strategy in control soybean *Phytophthora* root
rot this summer.

This strategy indeed will sound more effective if *Phytophthora* zoospores need to
absorb thiamine during mitigation. According to your kind suggestion, we checked the
thiamine utilization by *P. sojae* zoospores. The best method to determine the absorption
of thiamine is with isotopically labelled thiamine. However, we could not get ¹⁴C-labelled
thiamine. Thus, we measured the content of thiamine in zoospores by bioassay. 6×10^4
zoospores were added into the thiamine solution (0.1 nM), followed by incubation at
25°C for 0, 20 and 40 min. The zoospores were collected from 4 mL solution by
centrifugation at 7000 g. The cells were re-suspended in 800 μ L H₂O and disrupted by
Bioprep-24 Homogenizer (40 s, speed 6.00M/S, 3 times). 4 mL extracellular
supernatant was concentrated by freeze-drying and then dissolved in 800 μ L H₂O. The
thiamine content was determined by bioassay methods.

To verify the reliability of the method, we measured the detection limit of bioassay
 method with auxotrophic strain *E. coli* K-12 $\Delta thiE$. A linear relationship is identified
 between *E. coli* K-12 $\Delta thiE$ growth and thiamine content (Response Fig. 1a) with a
 detection limit of <0.05 nM. Meanwhile, we found that zoospores keep active migration
 and swimming in the solution during detection process. Thiamine content inside the
 zoospores increased at 20 min, indicating the transportation of thiamine from outside.
 When the incubation time last to 40 min, the thiamine inside the zoospores decreased.
 Thiamine concentration in the solution decreased all the time (Response Fig. 1b). These
 results indicate that swimming zoospores are able to absorb thiamine during swimming.
 Whereas, evidences with ^{14}C -labelled thiamine are needed to directly prove the
 thiamine utilization.

Response Fig. 1. Assay for thiamine utilization of *Phytophthora* zoospores during
 migration. **a**, Relationship between *E. coli* K-12 $\Delta thiE$ growth and thiamine content.

**b**, Determination of thiamine in and outside of *Phytophthora* zoospores during
 incubation of 0, 20 and 40 min. The data represent the means \pm SEM (n = 3). Means
 within columns followed by a different letter are significantly different ($p < 0.05$,
 ANOVA, Duncan's multiple range test).

1. Suggested Experiments.

The highest concentration of zoospores of *Phytophthora sojae* are produced by
 90 transferring several agar plugs to a fresh V8 agar plate. Repeated washing of the plate
 (every 20 min) is initiated two-three days later when most of the plate surface has almost
 covered the agar surface. Thiamine content of zoospores can be manipulated as follows.

Response: In our study, we prepared the zoospores by incubation of the mycelia
 of *Phytophthora* in sterile water. According to your kind suggestion, we preformed the
 supplementary experiments about the zoospores preparation by suggested method (PF
 Morris *et al.*, 1998), and 1.2×10^6 zoospores/plate (6-cm diameter) were obtained. By

calculating the biomass of *Phytophthora* mycelia from two methods, high quantity of
*P. sojae* zoospores are produced compared to liquid induction. Hence, zoospores used
in all supplementary experiments were prepared according to the recommended method.

The thiamine content of zoospores with high quantity (10^7 zoospores) was
analyzed by bioassay method, and thiamine stored within the spores was determined
with 1.32 ± 0.21 amol/zoospores, consisted with our obtained result (1.21 ± 0.38 amol/
zoospores). Otherwise, the lysate of zoospores with high concentration (2×10^7 cells/mL,
revised Fig. S5e) obviously improved the growth of strain K-12 $\Delta thiE$ compared to our
previous result (1×10^6 zoospores, Fig. S3e). The OD₆₀₀ value increased to 0.29 from
treatment of zoospores lysate compared to 0.12 in H₂O treatment (line 301-303).

Revised Fig. S5e Growth of strain K-12 $\Delta thiE$ in 96 well-plates with the addition
of thiamine ($1 \mu\text{M}$) and lysate of P6497 zoospores. The data represent the means \pm
SEM ($n = 3$). Means within columns followed by a different letter are significantly
different ($p < 0.05$, ANOVA, Duncan's multiple range test).

In the revised manuscript, Fig. S3e was replaced with the supplementary result
(present as Supplementary Fig. 5e in the revised manuscript) according to your
suggested experiments.

EGB colonization of hyphae producing sporangia may limit the production of zoosporangia
or the thiamine stores delivered to zoospores. Here I suggesting to test what happens
when EBG

**Response:** From our previous study (results present as Fig. S2d in the revised
manuscript), we observed that strain EGB cells adhered to the mycelia of *P. sojae* during
co-culture. According to your suggestion, we provided supplementary results to show

the production of zoospores during co-culture of EGB and *P. sojae*. Cells of strain EGB
adhered to mycelia remarkably limits the production of zoospores (presented as Fig.
S2e in the revised manuscript and below). These results showed that EGB adhering not
only inhibits *P. sojae* growth, but also eradicates zoospores production, both of which
are important for survival and plant infection of *Phytophthora*.

Revised Fig. S2e. Effect of strain EGB incubation on the production of
zoospores of *P. sojae* P6497. Scale bars: 100 μ m.

The result was added in the revised manuscript (line 177-178, Supplementary
material Fig. 2e).

In contrast does thiamine supplementation of zoospores improve their infectivity on
soybean leaves (Supplementary fig 3)?

Does thiamine supplementation diminish the effectiveness of EGB as a foliar biocontrol
agent?

**Response:** According to your kind suggestion, we conducted a supplementary
experiment regarding the effects of thiamine supplementation on *Phytophthora*
infection (mycelia and zoospores) and biocontrol efficiency of EGB.

Thiamine supplementation showed no obvious effects on the infection of the P6497
mycelia and zoospores from the *in vitro* leaf infection experiment. We deduced that the
thiamine content in leaves is high enough to support *Phytophthora* infection and growth,
because plants can synthesize thiamine (A Goyer *et al.*, 2010).

However, thiamine supplementation obviously decreased the biocontrol efficacy of
strain EGB and completely diminished inhibition effect of CcThi1, and increased the
lesion length of *P. sojae* on soybean leaves (present as Fig. S7c, d in the revised
manuscript, line 361-363). Otherwise, the effects of thiamine supplementation on
zoospores growth were also evaluated. We found that 2.96 μ M thiamine fully restores

the growth of P6497 in the presence of CcThi1 (present as Fig. S7a, b in the revised
 manuscript, line 357-361). These results indicated that thiaminase, but not other factor
 secreted by strain EGB inhibits oomycete growth.

These supplementary results and related descriptions have been provided in the
 revised manuscript (present as Supplementary Fig. 7).

Thank you for your kind suggestion, which is very important for the conclusion
 that EGB inhibits *Phytophthora* growth by thiaminase I-mediated thiamine deficiency.

 Revised Fig. S7. Effects of thiamine supplementation on zoospores growth and
 soybean infection of *Phytophthora*. **a, b**, Inhibition of mycelial growth on 10% V8
 plates by CcThi1 (0.24 nM) with supplementation of various concentrations of
 thiamine (0, 2.96×10^{-3} , 2.96×10^{-2} , 2.96×10^{-1} and $2.96 \mu\text{M}$) (**a**), and microscopic
 observation of mycelia after CcThi1 treatment was performed to measure the hyphal
 length after 7 d of incubation (**b**). **c, d**, Effects of thiamine supplementation on
 infections of *Phytophthora* mycelia and zoospores toward soybean leaves. All the
 leaves were stained with lactophenol trypan blue and decolorized with chloral hydrate
 (**c**), and the corresponding lesion diameters were measured (**d**). The data represent the

means \pm SEM (n = 5 or 10). Means within columns followed by a different letter are
significantly different ($p < 0.05$, ANOVA, Duncan's multiple range test).

2. Overall impression

A significant omission of the abstract was that there was no mention that the thiaminase
was excreted in vesicles.

**Response:** Thank you for your kind suggestion. We have revised the Abstract
based on our key findings with concise statement, and the OMVs-mediated secretion
of thiaminase I CcThi1 was also added (line 9-11).

In the introduction especially, I found the phrasing of ideas to be awkward,

**Response:** Based on the research status of metabolic exchange in plant holobionts,
we initially designed the Introduction into several sections: 1) Public goods mediated
microbial interactions is important for the ecological function of community; 2) Root
exudates serve as public goods and drive the assembly and functional maintenance of
the microbial community; 3) B-vitamins are public goods that essential for oomycetes
growth; 4) Predatory myxobacteria regulate the microbiome by predation or inhibition,
and thereby highlight our key findings. Our logic is that the thiamine mediated
microbial interaction occurred in plant holobiont. Since trophic interaction plays
important roles in shaping the microbiome, we put our research in the background of
plant microbiome. However, we realize that these descriptions are provided without
sufficient consideration of our scientific statement and we did not display the scenario
in microbiome level in this work.

According to your kind suggestion, we have reorganized the Introduction based
on the research background and key findings of the manuscript.

Part 1 (line 23-42): Public goods exchange is a crucial process driving the evolution of
microbial ecosystems;

Part 2 (line 43-59): Thiamine is regarded as public goods, consequences of thiamine
auxotrophy in *Phytophthora* pathogens are largely unclear in territorial ecosystem;

Part 3 (line 60-74): Root exudates-mediated root-associated bacterial communities
determine pathogen invasion, and exploring the thiamine supply and regulation in the
holobiont of *Phytophthora*-plant- rhizosphere microbes is meaningful.

Part 4 (line 75-89): Functions of predatory myxobacteria in microbial ecosystems
exerted by predation or inhibition were important, and thereby highlight out our core
findings.

I am not happy with the present title but with revisions, I still feel that this manuscript is a

very good fit for Nature Communications

**Response:** Your scientific comments has made us more aware of the biological
significance of CcThi1-mediated thiamine deficiency between myxobacteria and
*Phytophthora*. According to the key findings and importance, we revised the
manuscript title as: Myxobacteria restrain *Phytophthora* invasion by scavenging
thiamine in soybean rhizosphere via OMV-secreted thiaminase I.

3. Other comments:

Lines 180-183 the results are interpreted as “leading to the death of P sojae.

Lines 340-341 Later they point out that the mechanism is not killing but inhibition of growth
and also Fig. 2d

**Response:** In previous study, we preformed the co-cultural assay within 1 d, while
we realized that 1 d may be not long enough for myxobacteria to start predation. Hence,
we constructed the supplementary co-cultural assay of EGB and *P. sojae* within 3 d.

For the liquid co-cultural assay of *P. sojae* P6497 and EGB, the biomass of P6497
showed 25-fold decrease (Present as Fig. 2c and Supplementary Fig. 2c in the revised
manuscript), and abundant cells of strain EGB were observed to adhere to the mycelia
of P6497 (Supplementary Fig. S2d in the revised manuscript). Considering the cell wall
integrity of P6497 in EGB treatment, we deduced that the physical hindering by
myxobacterial cells limited substrates and oxygen transportation to P6497, which
indirectly leads to cell death.

In the later, we identified that myxobacteria EGB inhibited P6497 growth by
decomposing essential thiamine with a type I thiaminase CcThi1. By thiamine
supplement experiment and microscopic observation, we concluded that direct
interaction between EGB and P6497 is growth inhibition via thiamine competition, but
not cell killing.

I found the images of Fig.2a difficult to interpret. It would be better if the legend included a
sentence of explanation as I don't know what I am supposed to be seeing.

**Response:** For a clear presentation, revised Fig. 2a with illustrations of normal
(→) or inhibited (⊖) colony expansion of P6497 was provided, and the figure
instruction was also added in the revised legend.

For visualization of the *P. sojae* during co-culture, we used a GFP-labelled P6497
in Fig. 2a, similar results were obtained that EGB suppresses the growth of P6497
mycelia in a contact-independent manner. Meantime, we also identified that no

fluorescence quenching was detected from the co-culture assay on V8 plates, indicating
that EGB inhibits the growth of *P. sojae*, but not kills the cells.

In the revised manuscript, we moved the results of co-culture of EGB and P6497
to the Supplementary material as Fig. S2a, and the complementary co-culture result of
EGB and GFP-labelled P6497 was provided as revised Fig. 2a.

Thank you for your kind suggestion.

Lines 283-284...recombinant proteins against strain P6497, indicating that lipoation and
membrane association are not necessary for its growth-inhibitory effect against
*Phytophthora*

True but the efficiency of inhibition by direct contact supplementary 3F was much much
higher and should have been memorably mentioned in the discussion.

**Response:** Thank you for the helpful comment, which is important for the
exploring the experimental results in depth.

As we verified, CcThi1 could inhibit mycelial growth of *P. sojae* and reduce the
diameter of disease lesions in leaf infection experiments *in vitro*. In this experiment,
strain EGB was applied directly on the mycelial discs. Direct adhesion of EGB cells to
the mycelia allows myxobacteria to interact efficiently with *Phytophthora*, including
physical hampering on the viability and zoospore producing of *Phytophthora*, and the
high concentration of OMVs within local position. This may contribute to the improved
prevention of *Phytophthora* infection in the *in vitro* leaf infection experiment.

According to your kind suggestion, we have supplemented this discussion in the
revised manuscript (line 591-596).

Lines291 -294 germination of zoospores (Supplementary Fig. 3e). Furthermore, CcThi1
effectively limited the infection of soybean leaves by strain P6497 in the biocontrol assay.
The diameter of disease spots was significantly reduced with the increase of CcThi1
concentration (Supplementary Fig. 3f).

So is this negated by thiamin supplementation of swimming zoospores?

**Response:** As above mentioned, we carried out supplementary experiments
regarding the effects of thiamine supplementation on zoospores and mycelia infection
with or without presence of CcThi1 and EGB. Indeed, it was negated by thiamine
supplementation of swimming zoospores (Supplementary Fig. 7, line 357-363).

Entirely by accident I found possibly the earliest mention of the possible importance of
thiamine to phytophthora ERWIN DC, KATZNELSON H. Suppression and stimulation of
mycelial growth of *Phytophthora cryptogea* by certain thiamine-requiring and thiamine-

synthesizing bacteria. Can J Microbiol. 1961 Dec;7:945-50. doi: 10.1139/m61-119. PMID:
13890710. I only knew of the thiamin requirement from the text cited below.
My interest in zoospores within the oomycete research community is well known.
Combining that with the references that I felt needed to be cited is a clear indication of my
identity.

References

Chibucos MC, Morris PF. Levels of polyamines and kinetic characterization of their uptake
in the soybean pathogen *Phytophthora sojae*. Appl Environ Microbiol. 2006
May;72(5):3350-6. doi: 10.1128/AEM.72.5.3350-3356.2006. PMID: 16672477; PMCID:
PMC1472313.

D. C. Erwin and O. K. Ribeiro. 1996. *Phytophthora Diseases Worldwide* 592pp. St The
American Phytopathological Society, Paul, MN, USA:

Martinis J, Gas-Pascual E, Szydłowski N, Crèvecoeur M, Gisler A, Bürkle L, Fitzpatrick TB.
Long-Distance Transport of Thiamine (Vitamin B1) Is Concomitant with That of Polyamines.
Plant Physiol. 2016 May;171(1):542-53. doi: 10.1104/pp.16.00009. Epub 2016 Mar 22.
PMID: 27006489; PMCID: PMC4854701.

Mulangi V, Chibucos MC, Phuntumart V, Morris PF. Kinetic and phylogenetic analysis of
plant polyamine uptake transporters. Planta. 2012 Oct;236(4):1261-73. doi:
10.1007/s00425-012-1668-0. Epub 2012 Jun 19. PMID: 22711282.

**Response:** Thank you for your kind suggestion regarding to thiamine requirement
of *Phytophthora* and background of transporters. These references are crucial for us to
deeply understand the biological significance of the CcThi1-mediated thiamine
competition between myxobacteria and *Phytophthora*. The thiamine transportation in
plant will provide guidance for our future research on the absorption of thiamine in
*Phytophthora*.

The related references have been cited in the Introduction and Discussion to
highlight the importance of thiamine to *Phytophthora*.

Reviewer #3 (Remarks to the Author):

This impressive manuscript describes an extensive investigation into the importance of
thiamine for oomycete disease of soybean, and how thiaminases secreted by
myxobacteria can protect against disease by inhibiting oomycete growth.

This is a very important piece of work for the field. It is very thorough and provides a wealth
of data that deliver a mechanistic understanding of interactions between bacteria,

oomycetes and plants mediated by a specific metabolite. The findings are broadly relevant
to soil ecosystems, with potential applications in biocontrol and sustainable agriculture.
The manuscript is generally well-written, but would benefit from English language editing
as many sentences had minor grammatical problems.

**Response:** Thank you for the scientific evaluation for the importance of the
manuscript, which is important for our future related research.

The writing of manuscript has been polished by a professional English editing
agency to avoid grammatical problems, and we think the writing has been substantially
improved.

The taxonomy of the myxobacteria as presented in line 86 and around line 513 is wrong.
Myxobacteria were recently reclassified as a distinct phylum from Deltaproteobacteria
called the Myxococcota - see Waite et al (2020) and Oren and Garrity (2021), which also
affected the number of myxobacterial suborders.

**Response:** Thank you for the kind suggestion regarding the taxonomy of the
myxobacteria. These works (Waite DW *et al.*, 2020; Oren A *et al.*, 2021) provide new
insights for the evolutionary route of myxobacteria, which have brought extensive
attentions and discussions for us and myxo community when published.

Considering the importance of the reclassification (Waite DW *et al.*, 2020; Oren A
*et al.*, 2021), we have revised the distribution of thiaminase I in myxobacteria taxa
referring to the novel phylum-level lineages (phylum Myxococcota), and the
corresponding Fig. 6a has been revised. Thanks.

I found the introduction a bit confusing. It read like a list of statements about microbiomes
and there wasn't a clear logic running through it. Does the microbiome dictate which
metabolites are found in the public commons, or vice-versa? If public goods drive
microbiome resilience, how does mono-cropping render microbiome structure unstable,
and why is the accumulation of pathogens an indicator of instability?

**Response:** Metabolites from microbes and rhizosphere are commonly regarded as
public goods in ecosystems, and the metabolic cross-feeding is an important process
that broadly shape microbial communities and promote complex interactions between
microbes and plant (Mee M T *et al.*, 2014; Zhao M *et al.*, 2021). Recent researches
about rhizosphere show that root exudates dictate which health-promoting soil
microbes are found in the rhizosphere (Rolfe SA *et al.*, 2019). However, pathogens are
also able to utilize the public goods for growth by different strategies during co-
evolution (Gu S *et al.*, 2020). Hence, in relatively homogenous agricultural

monocultures, considerable spatial variation in disease dynamics exists (Wei Z *et al.*,
2018). We deduced that the resource depletion may change the microbiome structure,
resulting in the accumulation of pathogens or beneficial microbes. Since small initial
variation in soil microbiome composition and functioning can determine the outcomes
of plant-pathogen interactions (Wei Z *et al.*, 2019), accumulation of pathogens could
be regarded as indicator of instability. However, more evidences are needed.

Cooperation and competition commonly exist within microbial communities,
whereas there is limited understanding about the general principles guiding the
formation of these intricate systems. Your scientific suggestions “microbiome dictate
which metabolites are found in the public commons, or vice-versa” and “possible
contradiction between the public goods-mediated microbiome resilience and
microbiome structure unstable from mono-cropping” forced us to think how the public
goods-mediated interactions exist in a natural context.

Based on the research status of metabolic exchange between microbes and plant,
we initially designed the Introduction into several sections: Public goods mediated
microbial interactions is important for the ecological function of community; Root
exudates serve as public goods and drive the assembly and functional maintenance of
the microbial community; B-vitamins are public goods that essential for oomycetes
growth; Predatory myxobacteria regulate the microbiome by predation or inhibition,
and thereby highlight out core findings.

However, we realize that the descriptions are provided without sufficient
consideration of a clear logic. According to your kind suggestion, we have reorganized
the Introduction based on the research background and core experimental findings of
the manuscript.

According to your kind suggestion, we have reorganized the Introduction based
on the research background and key findings of the manuscript.

Part 1 (line 23-42): Public goods exchange is a crucial process driving the evolution of
microbial ecosystems;

Part 2 (line 43-59): Thiamine is regarded as public goods, consequences of thiamine
auxotrophy in *Phytophthora* pathogens are largely unclear in territorial ecosystem;

Part 3 (line 60-74): Root exudates-mediated root-associated bacterial communities
determine pathogen invasion, and exploring the thiamine supply and regulation in the
holobiont of *Phytophthora*-plant- rhizosphere microbes is meaningful.

Part 4 (line 75-89): Functions of predatory myxobacteria in microbial ecosystems

exerted by predation or inhibition were important, and thereby highlight out our core
findings.

I was also confused by the presentation of the earliest results sections, which seemed
unnecessarily complicated. Why use a bioassay for thiamine, when later it is detected
directly by UPLC-MS?

**Response:** As mentioned in Introduction section, B vitamins are common public
goods in oceanic algae communities. *Phytophthora*, which is phylogenetically related
to algae, is also known for the thiamine (vitamin B₁) auxotrophic property. However,
its thiamine resources in saprophytic stage is not clear in complex plant holobiont. We
thought it was necessary to clarify this question. In the Results section 3.1, we aimed
to identify if soybean root exudates and rhizosphere bacteria could provide thiamine for
growth of *Phytophthora*. The source of thiamine could determine the interactions
among soybean plant, *P. sojae*, and soil microbiome.

Since the detection limit (0.13 μ M) of HPLC is not low enough to measure
thiamine in the environments and medium, bioassay was used with *E. coli* K-12 Δ *thiE*
as the indicator (thiamine detection limit 0.05 nM). UPLC-MS was used to identify
thiamine decomposition product by SUPL, CcThi1 (1.19 μ M) and high concentration
of thiamine (0.29 mM) was incubated with thiaminase I at 25 °C for 6 h to detect the
product. It is much convenient and precise to measure thiamine by bioassay method.

For clear presentation, we have adjusted the presentation of Fig. 1 and added a
description (line 114-120) in the revised manuscript.

Why use strain T-1 to put thiamine into the system, when pure thiamine could have been
added directly? When using T-1 (Fig 1c), a better control than no *E. coli*, would have been
an *E. coli* which doesn't secrete thiamine.

**Response:** Here, our key point is to identify the thiamine supplier in rhizosphere.
We deduced that rhizosphere bacteria may provide thiamine for *Phytophthora*. Hence,
we recovered 1267 bacterial isolates from rhizosphere soil, and 65 isolates
demonstrated various degree of thiamine production. Among the 65 isolates, strain T-1
(Preliminary identified as *Escherichia* sp., not *E. coli*) was a higher thiamine producing
isolate and was selected as a representative of thiamine producers in soil. According to
the results (present as Fig. 1c, d in the revised manuscript), strain T-1 promoted the
growth of P6497, indicating that the thiamine exchange exists in plant holobiont
(present as Fig. 1e in the revised manuscript).

Meanwhile, pure thiamine was used as positive control to show that thiamine

producers support the growth of *Phytophthora* by supplying thiamine.

In Fig 1d, why is there as much growth with water for two of the species as there is with
thiamine – only one of the species seems to respond as expected to the water negative
control. In Fig 1b, it doesn't look like *P. sojae* is being stimulated by any of the additives –
it seems hard to reconcile the results in Fig 1b with the data in Fig 1d.

**Response:** The different *Phytophthora* growth responses to the additives is
partially attributed to their different growth rates and observed mycelia structures. *P.*
*sojae* mycelia expands much slower than *P. capsii* and *P. parasitica* when cultured in
V8 medium and tends to form compact colony, which results in different biomass under
same cultural condition.

During cultivation of P6497, weak growth of P6497 was observed in water
treatment. We deduced that stored thiamine within mycelia may support growth of
P6497 to some degree. However, the additives in the synthetic medium (P1 medium)
can support the increase the *Phytophthora* biomass with magnitude order. The
biomasses with additives were statistically higher than those in control (H₂O).

To further check the effects of additives on growth of *Phytophthora*, we repeated
the co-culture assay with optimal concentration of CaCl₂ (0.01%, m/v) in P1 medium
(Erwin DC *et al.*, 1961). Under this condition, growth of *P. sojae* was much evident.
*Phytophthora* DNA was isolated from a mixture of 3 wells (regarded as a repetition)
and biomass determined from three independent experiments (9 wells) by qPCR. The
raw data are provided in the files of Raw Source Data, and the related method and
results (present as Fig. 1a and b, Supplementary Fig. 1a and b in the revised manuscript)
are provided in the revised manuscript.

**Revised Fig. 1 Available thiamine from plant hosts and soil microorganisms promotes**
 **the growth of *Phytophthora*.**

The results showed that the additives promote the growth of *Phytophthora* in the
 synthetic medium (P1 medium, 0.01% CaCl₂), and the results in revised Fig 1a is
 reconciled with the data in revised Fig 1b.

In Fig 1c – how is the biomass of phytophthora distinguished from that of *E. coli*?

**Response:** *Phytophthora* biomass was quantified by qPCR using primers (PS-F1
 and PS-R1) specific for the actin gene (Table S3), which is not existed in genome of
 *Escherichia sp.* strain T-1.

I think it is difficult to claim that EGB's effect on *P. sojae* is not predatory, as the co-
 incubation experiments only seem to last 24h. I assume that predation is dismissed as
 there is no significant growth of EGB, but 24h is not a long time for myxobacteria to start
 showing signs of active predation – in our experience many predatory strains take 2-3 days
 to start exhibiting significant growth when preying upon susceptible organisms.

**Response:** Considering our previous studies, predation of bacteria and fungi (prey)
 by strain EGB was obviously observed after 1 d of incubation, hence, we tested the
 EGB's effect on *P. sojae* growth with 24 h of co-culture. However, as you suggested, 1
 442 d may be not long enough for myxobacteria to start predation.

Hence, according to your kind suggestion, we constructed the supplementary co-

cultural assay of EGB and *P. sojae* within 3 d. Results showed that co-cultures of EGB
 results in 25-fold decrease in P6497 biomass, but the biomass of EGB remains
 unchanged. These results indicated that strain EGB restrains the growth of *P. sojae* by
 inhibition rather than predation, and the adhering of EGB results in decrease of P6497
 biomass by indirect physical effect, which was consistent with the results after 1 d of
 incubation.

Fig. 2c and Supplementary Fig. 2c, Co-culture assay of P6497 (0.1 g, wet
 weight) and strain EGB (10⁵ cells/mL) or its secreted SUPL (10 mg/mL) in 25 mL of
 TV medium. The data represent the means ± SEM (n = 3). Means within columns
 followed by a different letter are significantly different (*p* < 0.05, ANOVA, Duncan's
 multiple range test).

Considering the importance of these results, the quantitative analysis of the
 biomass of strains EGB and P6497 from co-culture have been moved to the main text
 (Fig. 2c), and the growth state of co-culture was present as Supplementary Fig. 2c.
 Meantime, the SEM observation of the P6497 mycelia from the co-culture assay with
 3 d was also provided (revised Supplementary Fig. 2d). Related descriptions were also
 provided (line 172-175). Thanks.

Can the addition of thiamine stop the inhibitory effect of adding EGB? Line 203: Without
 showing that at this point in the paper it is difficult to conclude that it is specifically thiamine
 and not some other secretion of EGB that is inhibiting oomycete growth, which makes the
 conclusion in line 213-214 difficult to justify.

**Response:** According to your kind suggestion, we conducted a supplementary
experiment regarding the effects of thiamine supplementation on *Phytophthora*
infection (mycelia and zoospores) and biocontrol efficiency of EGB.

Thiamine supplementation showed no obvious effects on the infection of the P6497
mycelia and zoospores from the *in vitro* leaf infection experiment. We deduced that the
thiamine content in leaves is high enough to support *Phytophthora* infection and growth,
because plants can synthesize thiamine (A Goyer *et al.*, 2010).

However, thiamine supplementation obviously decreased the biocontrol efficacy of
strain EGB and completely diminished inhibition effect of CcThi1, and increased the
lesion length of *P. sojae* on soybean leaves (present as Fig. S7c, d in the revised
manuscript, line 361-363). Otherwise, the effects of thiamine supplementation on
zoospores growth were also evaluated. We found that 2.96 μ M thiamine fully restored
the growth of P6497 in the presence of CcThi1 (present as Fig. S7a, b in the revised
manuscript, line 357-361). These results indicated that thiaminase, but not other factor
secreted by strain EGB inhibits oomycete growth.

These supplementary results and related descriptions have been provided in the
revised manuscript (present as Supplementary Fig. 7).

Thank you for your kind suggestion, which is very important for the conclusion
that EGB inhibits *Phytophthora* growth by thiaminase I-mediated thiamine deficiency.

Revised Fig. S7. Effects of thiamine supplementation on zoospores growth and soybean infection of *Phytophthora*. a, b, Inhibition of mycelial growth on 10% V8 plates by CcThi1 (0.24 nM) with supplementation of various concentrations of thiamine (0, 2.96×10^{-3} , 2.96×10^{-2} , 2.96×10^{-1} and $2.96 \mu\text{M}$) (a), and microscopic observation of mycelia after CcThi1 treatment was performed to measure the hyphal length after 7 d of incubation (b). c, d, Effects of thiamine supplementation on infections of *Phytophthora* mycelia and zoospores toward soybean leaves. All the leaves were stained with lactophenol trypan blue and decolorized with chloral hydrate (c), and the corresponding lesion diameters were measured (d). The data represent the means \pm SEM ($n = 5$ or 10). Means within columns followed by a different letter are significantly different ($p < 0.05$, ANOVA, Duncan's multiple range test).

What is the significance of thiaminase being secreted in OMVs? If it is periplasmic in the cell, would it not be in the lumen of the OMV? How could it access thiamine substrate? Is the OMV permeable to thiamine, or do the OMVs lyse to release the enzyme?

**Response:** Commonly, enzymes involved in extracellular macromolecules
decomposition are entrapped on cell surface by lipo, while counterparts in Gram
negative bacteria are secreted in periplasm. Such enzymes are retained closely to the
cells by these approaches for effective nutrient acquisition. To our surprise, thiaminase
I in Gram negative myxobacteria is generally lipoation and outer membrane located as
revealed by signal peptides structure analysis. Gram negative bacteria have long been
found producing outer membrane vesicles (OMVs). OMVs play roles in pathogenesis,
cell-to-cell communication, and stress responses (Sartorio *et al.*, 2021). OMVs have
been considered playing roles in killing prey cells at distance (Whitworth *et al.*, 2011;
Evans *et al.*, 2012), which translocate hydrolases or antimicrobial substances to the
target cells. Here, we deduce that CcThiI could disperse in the environment via OMVs
blebbing. Otherwise, locating on OMVs might increase the stability of CcThiI in harsh
environment, and also allows myxobacteria to interact with *Phytophthora* by promoting
adhesion to the cell surfaces. However, its biological significance needs further
investigation.

We failed to discriminate its location on OMVs by protease digestion for its
protease resistance property. However, intact OMVs and broken OMVs by
ultrasonification showed similar thiaminase I activity (Response Fig. 1). No specific
transporter has ever been identified on outer membrane, we deduced that OMVs is
permeable to thiamine. Thus, its location on the surface or in the lumen may not restrict
its enzymatic activity. Anyway, we think your comments raised interesting topics on
the structure of bacterial OMVs. We will try to determine its location on the OMVs in
our future work.

**Response Fig. 1** Measurement of thiaminase activity of OMVs and ultrasonic
OMVs. The data represent the means \pm SEM ($n = 3$). Means within columns followed
by a different letter are significantly different ($p < 0.05$, ANOVA, Duncan's multiple

range test).

Discussion regarding the significance of thiaminase secretion via OMVs was
supplemented in the revised manuscript (line 586-596). Thanks.

Are myxobacteria generally prototrophic for thiamine?

**Response:** Many bacteria can synthesize thiamine, while thiamine auxotrophs
must obtain it or its precursors from the environment. Commonly, the pyrimidine
moiety is derived from the purine intermediate 5-aminoimidazole ribonucleotide (AIR),
which is converted to hydroxymethyl pyrimidine phosphate (HMP-P) by ThiC (HMP
synthase). Then, HMP-P is phosphorylated to HMP-PP by ThiD (kinase). The G3P and
glycine could be combined to synthesize HET-P in a multistep process by Dxs (1-
deoxyd-d-xylose-5-phosphate synthase), ThiO (glycine oxidase) and ThiG (thiazole
synthase). Thiamine phosphate synthase (ThiE) combines HET-P and HMP-PP to form
thiamine monophosphate (TMP), and typically ThiL phosphorylates TMP to produce
the active cofactor TPP (Response Fig. 2a, below, Sannino DR *et al.*, 2018).

The strains *M. xanthus* DK1622 and EGB used in this research are thiamine
prototrophs. Genome analysis of myxobacteria with the available genome sequences
showed that the key enzymes ThiC, ThiD, thiG and thiE and homologs are widely
distributed in most members of myxobacteria taxa (2 classes, 4 orders, 7 families and
18 genera) (Response Fig. 2b, below,), indicating that myxobacteria are generally
prototrophic for thiamine.

To further verify this conclusion, we deleted the gene encoding ThiC in the model
strain *M. xanthus* DK1622, generating strain CL1006. Strain DK1622 grew on CTT-1
medium (no thiamine), and addition of thiamine promoted its growth. However, *thiC*
mutant strain CL1006 lost the ability to grow on CTT-1 medium, which could be
partially restored by thiamine addition (Response Fig. 2c). All these results show that
myxobacteria are generally prototrophic for thiamine and reserve the ability to utilize
exogenous thiamine.

However, considering the aim of the manuscript, these results were not presented
with in-depth discussion in the revised manuscript.

**Response Fig. 2. Mycobacteria are generally prototrophic for thiamine. a, c, Predicted**
 **thiamine biosynthesis pathway in mycobacteria referring to *Burkholderia* (Sannino**
 **DR *et al.*, 2018); b, Distribution of ThiC, ThiD, thiG and thiE homologs in**
 **mycobacterial taxa. c, Effects of thiamine on growth of DK1622 and *thiC* mutant**
 **strain CL1006 on CTT-1 medium. The data represent the means ± SEM (n = 3).**

**** $P < 0.005$, **** $P < 0.0001$, Student's t-test.**

**Minor comments**

**Line 91 – in what way is it novel?**

**Response:** The function of CcThi1 homologs in mycobacterial genomes is not
 annotated before our work. It was conceptually annotated as a solute-binding protein.
 Sequence analysis showed that CcThi1 from strain EGB shared rather low sequence
 identity (less than 20%) with known type-I thiaminases, and phylogenetic tree analysis
 showed that thiaminase I enzymes from mycobacteria formed a separate branch from
 those of other species, indicating its novelty on sequence level. Till now, all reported
 thiamine I are extracellular protein secreted into the environment, however, most
 homologs of CcThi1 in mycobacteria are predicted to be outer membrane located as a

lipoprotein. Furthermore, thiaminase-mediated metabolite interaction between
myxobacteria and *Phytophthora* provides new insights for the ecological significance
of thiamine, indicating functional novelty. Hence, we concluded that CcThi1 from
myxobacteria is novel thiaminase that mediated new interaction of myxobacteria and
*Phytophthora*.

Line 92 – it is not strictly increasing the plant's resistance to *Phytophthora*, but
reducing the amount of pathogen.

**Response:** According to your suggestion, we have reorganized the Introduction
based on the research background and our key findings. The related description has
been revised as: Myxobacteria release a novel thiaminase to scavenge the public
thiamine in the soil, which in turn regulate the thiamine content and promote plant
protection (line 83-85). Thanks.

Line 95 – The statement about HGT seems throwaway. What do you mean by this
statement? How is it relevant?

**Response:** Previous research show that oomycetes feed by secreting
depolymerizing enzymes (public goods) to process complex substrates in the
extracellular environment, and taking up the resulting simple nutrients into their own
cells (Richards TA *et al.*, 2013). During the evolutionary process, horizontal gene
transfer (HGT) may play major role in in shaping the repertoire of these enzymes.

However, this introduction was not closely relevant to our findings. According to
your kind suggestion, we have deleted these statements and reorganized the part of
Introduction. Thanks.

Line 96-97. In what way is it unique? And what do you mean by fine-tuned? Fine-tuned
implies that the concentration of thiamine and thiaminase in soil are tightly-regulated, but
that hasn't been shown – rather that small changes in thiamine can have substantial
impacts on oomycete growth and pathogenesis.

**Response:** Public goods in microbial ecosystem have been shown to regulate the
relationships of microbial members. Microbes engage in the acquisition and protection
of public goods by diversified strategies, including resource limitation and cell killing,
such as siderophore–iron complexes in iron acquisition (Cordero O X *et al.*, 2012) and
cell killing for exploiting public goods (Loneragan Z R *et al.*, 2019). Consequently,
social behaviors such as cooperation, cheating or division of labor arise in microbial
communities. However, mechanisms involved in regulation of public goods supplying
such as thiamine in microbial communities remain underexplored.

Supplying of thiamine in *Phytophthora*-soil microorganism-soybean system
offered an ideal model for investigating public good regulation. Here, we hypothesize
that thiamine content might be regulated in plant holobiont. Our results showed that
CcThi1 played a role in regulating soil thiamine concentration. Public goods supplier
protects its public goods by means mentioned above. Here, we proved that public goods
like thiamine is regulated in ecosystem by the third party (myxobacteria) after released
from producer. We think this mechanism is unique in microbiome.

From the pot experiment (Fig. 5), we speculated that the occurrence of the
*phytophthora* disease may be related to the thiamine content. To test this speculation,
we further determined the biomass of *P. sojae* and the thiamine content in the soil from
different treatments. We identified positive correlations between the thiamine
concentration and disease index or *P. sojae* biomass (Fig. 5d-g). Indeed, as you
mentioned, our results showed that small changes in thiamine indeed have substantial
impacts on oomycete growth and pathogenesis (Fig. 5c-g). However, to what extent
myxobacteria fine-tuned the availability of public goods in microbial communities is
not verified. More evidence of the fine-tuned mechanism is needed. Hence, according
to our present results, we have revised the related description as: Our experimental
findings highlight a unique trophic interaction by intervening the availability of public
goods in microbial communities (line 85-86). Thanks.

Line 172 – words like ‘plunder’ and ‘greedily’ seem excessively poetic/emotive. (also
‘pivotal’ line 235, ‘loopholes’ line 557 and ‘raiding’ – several places including abstract).

**Response:** Some descriptions like “plunder, greedily, pivotal, loopholes, et.” have
been revised for scientific presentation. Thanks.

Line 178 – why would suppression of *P sojae* growth be unexpected when EGB is a known
predator?

**Response:** As opposed to fungi, the major composition of oomycetes cell walls
consists essentially of cellulose and β -1,3-glucans, while chitin and β -1,6-glucans occur
in small amounts in the cell walls of some oomycetes. In our previous research, we
found that *Corallococcus* sp. EGB preyed on fungi by cell wall degrading enzyme (β -
1,6-glucanase), and predated *Phytophthora* by *Archangium* sp. AC19 via specialized
CAZyme system, Considering the content of cellulose and β -1,3-glucan in cell wall of
*Phytophthora*, we had thought that strain EGB may have the ability to predate *P. sojae*.
However, our results negated this opinion, since strain EGB could not grow with
*Phytophthora* biomass as resource.

To avoid confusion, we have revised the manuscript to remove such emotional
expression in scientific presentation (line 167-170). Thank you for your kind suggestion.
Fig 2 – I found it difficult to relate panel b to panel a. Panel c didn't seem to add much to
the manuscript. In panel d, rather than comparing with/without EGB extract, it would have
been more interesting to see what the mycelia looked like at the edge of where growth was
being inhibited by EGB in the experiments of panel a.

**Response:** For a clear presentation, revised Fig. 2a with illustrations of normal
(→) or inhibited (⊥) colony expansion of P6497 was provided, and the figure
instruction was also added in the revised legend. The colony radius of normal (→) or
inhibited (⊥) growth of *P. sojae* was measured (revised Fig. 2a, b). By MTT assay, we
identified that SUPL-treated mycelia are metabolically active, combining the results that
SUPL treatment exhibits no fluorescence quenching, we concluded that secreted SUPL
from strain EGB inhibited growth of *Phytophthora*, but not killed the mycelia. Hence,
the measurement of metabolic activity was kept in the text and present as Fig. 2d in the
revised manuscript, while the observation of MTT staining of P6497 mycelia was
moved into supplementary material as Fig. S4a.

Otherwise, for visualization of the *P. sojae* during co-culture, we used a GFP-
labelled P6497 in Fig. 2a, similar results were obtained that EGB suppresses the growth
of P6497 mycelia in a contact-independent manner. Meantime, we also identified that
no fluorescence quenching was detected from the co-culture assay on V8 plates,
indicating that EGB inhibits the growth of *P. sojae*, but not kills the cells.

In the revised manuscript, we moved the results of co-culture of EGB and P6497
to the Supplementary material as Fig. S2a, and the complementary co-culture result of
EGB and GFP-labelled P6497 was provided as revised Fig. 2a (below). The results of
microscopic analysis of SUPL-treated GFP-labelled P6497 mycelia have been moved
to Supplementary material as Fig. S4b.

Revised Fig. 2a. Co-culture assay of strains EGB and GFP-labelled P6497 on
 10% V8 plates with (EGB// P6497) or without (EGB+P6497) membrane separation.

The blue dotted line indicates the semi-permeable membrane (7 kDa molecular
 weight cutoff), the red dotted line indicates the membrane filter (0.22 μm pore size),
 dotted arrow indicates the location of strain EGB; → indicates the normal growth, ⊥
 indicates the inhibited growth. Scale bar, 1 mm.

Line 280. What do you mean by 'functional confirmation'? At this point in the manuscript
 the gene's function has not been confirmed.

**Response:** The description 'functional confirmation' mean that the purified
 protein is originally annotated as an extracellular solute-binding protein in the genome,
 which was experimentally identified as thiaminase I enzyme in our work. However, as
 you mentioned, the gene's function was confirmed by heterologous expression and
 activity determination in the following study, which was not verified at this point.

The statement has been moved to line 314-315 in the revised manuscript to avoid
 of logic confusion. Thanks.

Line 300. It would be more informative to present K_m values rather than specific activities,
 as K_m can be related to ecologically relevant concentrations of substrate.

**Response:** According to your kind suggestion, the K_m values were calculated and
 provided in the revised manuscript based on calculation of the raw data, and the result
 and some description was also added (Supplementary Fig. 5d in the revised manuscript,
 line 309-310).

Line 491 – the authors state that thiaminase is predominantly (86%) in OMVs, but the

western blot in Fig 4 seems to suggest otherwise.

**Response:** The statement “85.6% of the hydrolytic activity was concentrated in
OMVs” meant that, of the secreted CcThiI in supernatant (SUP), 85.6% thiaminase I
activity was in OMVs fraction, residual 14.4% activity was in SUPU (OMV-free SUP)
(Fig. 4b). In Fig. 4c, the western blot analysis was performed using the isolated
membrane fraction and OMVs from myxobacterial strains, which showed that CcThiI
is located on membrane and OMVs.

To avoid the misunderstanding, we provided the supplementary western blot of
the SUP, SUPU (OMV-free SUP) and isolated OMVs from SUP, which are consistent
with the fractions that were used in activity measurements (presented as Fig. 4d in the
revised manuscript). The western blot analysis showed that CcThiI was located on
membrane and SUP. In OMV-free SUP fraction (SUPU) obtained after
ultracentrifugation, CcThiI content was significantly lower, indicating that thiaminase
I CcThiI from SUP was mainly located on OMVs (Fig. 4d in the revised manuscript).

The supplementary result has been added in the revised manuscript along with the
presentation of activity detection (Fig. 4d, line 431-433).

Fig. 4d Western blot analysis of CcThiI in SUP, SUPU (OMV-free SUP) and
isolated OMVs from SUP using the anti-CcThiI antibody.

Line 610-612. I didn't understand this sentence or it's significance.

**Response:** Thiamine synthesis is an energy-intensive biochemical process.
Organisms often tend to use exogenous thiamine for growth and reproduction to save
energy. This process is usually controlled by riboswitch (McRose D *et al.*, 2014).
Riboswitch played roles in cells at μM concentration level of thiamine. As shown in
our results, thiamine concentration in rhizosphere is at pM level. Under such condition,
riboswitch could not work, instead, by possible public thiamine regulation.

However, we realized that this statement was not direct relevant to our key findings.
We have reorganized this statement and other parts of Discussions, including the
significance of OMVs (line 586-596), thiamine regulation (line 626-629).

Thank you for your kind suggestion, which is important for us to in-depth
understand the experimental findings.

Reviewers' Comments:

Reviewer #2:

Remarks to the Author:

I recommend acceptance of the revised manuscript.

Reviewer #3:

Remarks to the Author:

I'd like to thank the authors for responding to my comments on their original manuscript. I'm very happy with the lengths the authors have gone to, including the additional experiments they have performed, which have more than satisfactorily addressed all my queries - thank you.